# NEURO-MoBRE: EXPLORING MULTI-SUBJECT MULTI-TASK INTRACRANIAL DECODING VIA EXPLICIT HETEROGENEITY RESOLVING

## ABSTRACT

Neurophysiological decoding, fundamental to advancing brain-computer interface technologies, has significantly benefited from recent advances in deep learning. However, existing decoding approaches largely remain constrained to single-task scenarios and individual subjects, limiting their broader applicability and generalizability. Efforts towards creating large-scale neurophysiological foundation models have shown promise, but continue to struggle with significant challenges due to pervasive data heterogeneity across subjects and decoding tasks. Simply increasing model parameters and dataset size without explicitly addressing this heterogeneity fails to replicate the scaling successes seen in natural language processing. Here, we introduce the **Neural Mixture of Brain Regional Experts** (**Neuro-MoBRE**), a general-purpose decoding framework explicitly designed to manage the ubiquitous data heterogeneity in neurophysiological modeling. Neuro-MoBRE incorporates a brain-regional-temporal embedding mechanism combined with a mixture-of-experts approach, assigning neural signals from distinct brain regions to specialized regional experts on a unified embedding basis, thus explicitly resolving both structural and functional heterogeneity. Additionally, our region-masked autoencoding pre-training strategy further enhances representational consistency among subjects, complemented by a task-disentangled information aggregation method tailored to effectively handle task-specific neural variations. Evaluations on intracranial recordings from 11 subjects across five diverse tasks, including complex language decoding and epilepsy diagnosis, demonstrate that Neuro-MoBRE surpasses prior art and exhibits robust generalization for cross-subject decoding on unseen subjects.

## 1 INTRODUCTION

Neurophysiological signals represent the direct electrical activity generated by the central and peripheral nervous systems, as characterized by the volume conduction theory (Nunez & Srinivasan, 2006). With advancements in biomedical engineering and continuous progress in neuroscience, neurophysiological decoding has increasingly become a critical focus of research, especially in the development of BCIs. Simultaneously, breakthroughs in deep learning, have significantly enhanced the decoding capabilities of BCIs, enabling sophisticated interpretations of human sensory, motor, and communicative functions through implantable technologies (Robinson et al., 2024; Bouton et al., 2016; Metzger et al., 2023; Willett et al., 2023; Hochberg et al., 2012; Benabid et al., 2019). Nonetheless, current methods remain predominantly confined to task-specific decoding scenarios and restricted datasets gathered from individual subjects, thereby excelling only in narrowly defined contexts. This limitation severely constrains their practical utility and generalizability across diverse neurophysiological contexts.

Recent studies (Yang et al., 2023; Zhang et al., 2024; Wang et al., 2023) have explored large-scale neurophysiological foundation models by employing various pre-training techniques with electroencephalogram (EEG) data aggregated from multiple tasks. These approaches aim to replicate the scalability benefits observed in large language models (LLMs) (Kaplan et al., 2020). Techniques such as contrastive learning, masked autoencoding, and vector quantization have been adopted to extract generalized neural representations from extensive datasets (Jiang et al., 2024; Wu et al., 2025). Nevertheless, such methods still require task-specific fine-tuning, restricting their general-

purpose decoding potential. Recently, NeuroLM (Jiang et al., 2025) introduced a multi-task EEG decoding model, framing diverse decoding tasks as a universal question-answering framework via pre-trained LLMs. Despite pioneering multi-task EEG decoding, NeuroLM exhibits substantially lower performance compared to specialized single-task models.

Despite these efforts, the development of large neurophysiological models (LNMs) capable of robust general-purpose decoding remains elusive due to several unresolved challenges:

(1) **Ubiquitous data heterogeneity in neurophysiological modeling.** Data heterogeneity encapsulates the variability and inconsistency inherent in neurophysiological recordings, arising from biological differences among subjects as well as variations in experimental setups and conditions. For example, neurophysiological signals significantly differ across task paradigms due to distinct neural processes engaged in task-specific recruitment. Current approaches typically lack explicit strategies for addressing data heterogeneity, often limiting themselves to structural adjustments at the model architecture level, such as accommodating differences in channel numbers and signal duration, neglecting effective mechanisms to address neurophysiological variations across subjects and neural activity contexts. Consequently, large-scale modeling of heterogeneous data encounters significant consistency issues, resulting in suboptimal decoding performance.

(2) **Limited spatiotemporal resolution in non-invasive recordings.** Non-invasive modalities such as EEG suffer from signal attenuation caused by the skull and dura mater, severely constraining their capacity to capture high-resolution neural dynamics. While existing LNMs can decode neural activity for relatively coarse tasks, such as abnormal event detection, they fall short for cognitively complex tasks, including fine-grained somatic motor control and language decoding, where access to high-frequency components (e.g., high-gamma bands) is critical. This limitation is compounded by the field's reliance on heterogeneous, publicly available datasets collected with diverse protocols, equipment, and subject populations, further exacerbating variability. Together, these factors hinder the development of LNMs capable of consistent, multi-subject, multi-task decoding across domains of high cognitive and motor complexity.

To effectively address the data heterogeneity challenge, we propose a general-purpose neurophysiological decoding framework termed **Neural Mixture of Brain Regional Experts** (**Neuro-MoBRE**). Neuro-MoBRE simultaneously performs multi-task decoding of neurophysiological signals from intracranial recordings across multiple subjects. Our fundamental strategy is straightforward: **explicitly resolving consistency conflicts arising from inherent data heterogeneity to achieve unified and robust neural decoding.** Specifically, Neuro-MoBRE incorporates a brain-regional-temporal embedding mechanism within a decoder-only transformer architecture to effectively handle structural heterogeneity. Additionally, it utilizes a mixture-of-experts approach, assigning neural signals from distinct brain regions and individual subjects to specialized regional experts, thus explicitly addressing both task-induced and subject-induced heterogeneity. We further introduce a region-masked autoencoding pre-training scheme designed to unify parameters from independently trained subject-specific models, serving as an effective initialization step for subsequent multi-task and multi-subject decoding. This mechanism significantly enhances representational homogeneity among subjects. Recognizing that distinct tasks activate different brain regions and neural pathways, we also propose a task-disentangled information aggregation method to integrate neural features extracted by various regional experts, thereby further mitigating task-specific heterogeneity and boosting decoding performance.

To circumvent the limitations posed by limited spatiotemporal resolution in non-invasive neurophysiological recordings, we rigorously evaluate Neuro-MoBRE using intracranial data collected from 11 subjects across five distinct decoding tasks. Unlike prior studies focusing primarily on binary or simple multi-class classification tasks, our evaluations span complex decoding scenarios, including language decoding and epileptic seizure diagnosis. Extensive experimental results demonstrate that Neuro-MoBRE surpasses traditional single-task decoding methods and existing large-scale neurophysiological foundation models. Furthermore, Neuro-MoBRE exhibits robust generalization in cross-subject decoding for unseen subjects. Collectively, our contributions are as follows:

(1) We introduce Neuro-MoBRE, a versatile general-purpose decoding framework explicitly designed to resolve inherent data heterogeneity in neurophysiological signals, enabling effective multi-subject and multi-task decoding;

(2) To facilitate the effective extraction of meaningful information required to manage data heterogeneity, we conduct comprehensive evaluations using intracranial recordings from a diverse cohort of 11 subjects across complex tasks, including phoneme articulation and seizure diagnosis;

(3) Our extensive experimental results validate the superior performance of Neuro-MoBRE, achieving an average top-1 decoding accuracy improvement of 17.98% compared to existing multi-task approaches. Additionally, Neuro-MoBRE demonstrates robust cross-subject generalization capabilities to previously unseen subjects.

## 2 RELATED WORK

**Intracranial Neurophysiological Decoding.** Intracranial brain recordings, including electrocorticography (ECoG) and stereo-electroencephalography (sEEG), offer unparalleled spatial and temporal resolution for investigating neural substrates of motor, sensory, and cognitive functions (Bouton et al., 2016; Benabid et al., 2019; Hochberg et al., 2012; Willett et al., 2023; Metzger et al., 2023). Recent work by Hochberg et al. (Hochberg et al., 2012) demonstrated that individuals with tetraplegia could perform complex robotic arm tasks using signals from implanted microelectrode arrays. Likewise, advances in speech prosthesis show high-accuracy text or audio decoding from attempted speech through intracranial recordings (Willett et al., 2023; Metzger et al., 2023; Moses et al., 2021; Angrick et al., 2019), including systems achieving low word error rates using signals from ventral premotor and Broca's areas (Willett et al., 2023). However, prior studies largely focus on single-task, single-subject scenarios, limiting generalizability. Our work addresses these constraints by integrating multi-task, multi-subject intracranial recordings to enable robust and generalizable decoding.

**Large-scale Neurophysiological Models.** Successes of LLMs (Brown et al., 2020) have inspired scalable brain decoding models that integrate signals from many subjects and tasks. MMM (Yi et al., 2023) standardizes EEG across sensor grids via topology-agnostic pretraining; BIOT (Yang et al., 2023) unifies biosignal representations using a handcrafted tokenizer. Brant (Zhang et al., 2024) and BrainBERT (Wang et al., 2023) both use masked autoencoding in the frequency domain for intracranial data, while LaBraM (Jiang et al., 2024) leverages discrete EEG tokenization. However, these methods require fine-tuning per task and lack true multi-task capability. NeuroLM (Jiang et al., 2025) reframes decoding as question answering for multi-task use, while H2DiLR (Wu et al., 2025) explicitly disentangles heterogeneity, though limited to single-task scenarios. Our framework extends this line with explicit modeling of both task- and subject-level heterogeneity for robust multi-task decoding; further related work details are provided in the Appendix A.

## 3 METHODOLOGY

Neuro-MoBRE consists of three key components designed to explicitly address the inherent data heterogeneity: a brain-regional-temporal tokenizer, a brain-regional mixture-of-experts module, and a task-disentangled information aggregation mechanism, each tailored to mitigate different aspects of heterogeneity (Fig. 1). We begin by introducing the overall decoding paradigm of Neuro-MoBRE in Sec. 3.1. Furthermore, in Sec. 3.2, we introduce a region-masked autoencoding pre-training (RMAE) strategy that leverages independently trained subject-specific models to initialize Neuro-MoBRE, effectively upcycling learned parameters for further heterogeneity resolving.

### 3.1 NEURO-MOBRE OVERVIEW

**Problem Formulation.** Suppose a set of neurophysiological recordings $\mathcal{X}$ are collected from $m$ subjects $\{\mathcal{S}_i\}_{i=1}^{m}$, across a set of $n$ decoding tasks $\{\mathcal{T}_j\}_{j=1}^{n}$. Each recording $\mathcal{X}_i \in \mathbb{R}^{N_i \times T_i \times C_i}$ consists of $N_i$ data samples, each with a signal length of $T_i$ and $C_i$ channels for the subject $\mathcal{S}_i$ with substantial heterogeneity introduced by both neurophysiological variability across subjects and differences in experimental protocols across tasks. Neuro-MoBRE aims to achieve robust decoding in a multi-subject, multi-task setting by learning a mapping function $f(\cdot)$ that projects the input recordings to their corresponding task-specific labels, i.e., $f(\mathcal{X}) \rightarrow \mathcal{Y}_{\mathcal{T}}$ for all decoding tasks.

**Brain-regional-temporal Tokenizer.** Since intracranial neurophysiological recordings from different subjects typically originate from diverse brain regions, each region potentially having a varying

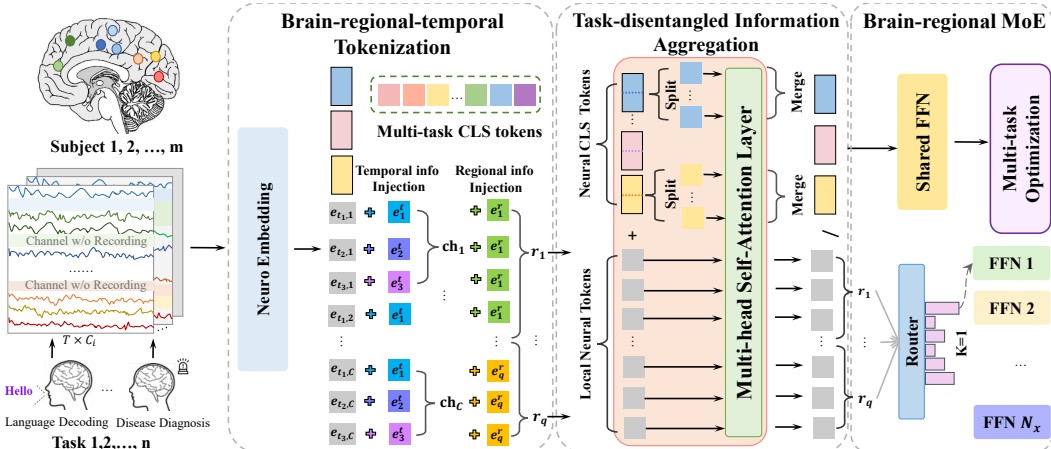

Figure 1: Overview of Neuro-MoBRE for multi-subject, multi-task neural decoding. The brain-regional-temporal tokenizer resolves structural heterogeneity by projecting neural signals collected from diverse recording configurations into a unified representation space, incorporating dedicated temporal and regional tokens ($\mathbf{e}^t$, $\mathbf{e}^r$). Channels and brain regions are denoted by ch and $r$, respectively. Disentangled "wider" neural CLS tokens aggregate local neural features and enhance global neural representation capacity, capturing diverse functional patterns across distinct cognitive tasks. To further mitigate functional heterogeneity among brain regions and tasks, all CLS tokens share a common FFN, while local tokens are dynamically routed to their associated regional expert networks.

channel count, structural heterogeneity naturally arises. To resolve this, we propose embedding the recordings in a regional-temporal manner. Specifically, for each one of the $N_i$ neuro recordings $X \in \mathbb{R}^{T \times C}$ from recording $\mathcal{X}_i$, we employ a cascade of 1-D convolutional operations with receptive field size $k$ over the temporal dimension, transforming the input into a sequence of embeddings $E = [\mathbf{e}_{p,c} \in \mathbb{R}^d \mid p = 1, \ldots, \lfloor \frac{T}{k} \rfloor; c = 1, \ldots, C]$, where each $\mathbf{e}_{p,c}$ represents the $p^{\text{th}}$ temporal token corresponding to channel $c$. To explicitly capture temporal information, we augment these embeddings with learnable temporal positional embeddings $[\mathbf{e}_p^t \mid p = 1, \ldots, \lfloor \frac{T}{k} \rfloor]$ for each channel, resulting in the temporally-aware embeddings $[\mathbf{e}_{p,c} + \mathbf{e}_p^t \mid p = 1, \ldots, \lfloor \frac{T}{k} \rfloor; c = 1, ..., C]$. Furthermore, as the intracranial channel configuration does not adhere to a standardized arrangement (such as the EEG 10-20 system), we explicitly inject regional information into each embedding to encode brain region specificity. A set of brain regions $\mathcal{R}_j$ is considered relevant to a decoding task $\mathcal{T}_j$. We define a set of learnable region-specific embeddings $[\mathbf{e}_q^r \mid q = 1, \ldots, |\mathcal{R}_j|]$. For each token $\mathbf{e}_{p,c}$, we integrate regional information by adding the embedding $\mathbf{e}_q^r$ corresponding to the region $q$ from which channel $c$ was recorded:

$$\mathbf{e}_{p,c}^{\text{final}} = \mathbf{e}_{p,c} + \mathbf{e}_p^t + \mathbf{e}_q^r. \tag{1}$$

**Brain-regional MoE Block.** The architecture of the proposed Neuro-MoBRE follows a typical decoder-only Transformer with layers, each containing a multi-head self-attention (MSA) structure with pre-normalization and a feed-forward network (FFN) with residual connection. To resolve data heterogeniety casued by brain region inconsistency due to brain region involvement and neuro-physiological differentces, we replace the FFNs of each layer with a Brain-regional MoE module, which comprises a router and several experts, $\{\text{FFN}_x\}$. In this work, we specifically set the number of experts $N_x = |\{\text{FFN}_x\}|$ equal to the number of brain regions of interest. The rationale behind aligning the number of experts with brain regions is based on the assumption that neural activities across different brain regions, and particularly across different decoding tasks, vary significantly. Thus, tokens corresponding to different brain regions should ideally be processed by specialized experts. The router dynamically determines the assignment of each token to the most suitable expert, allowing each expert network to specialize in capturing distinct neural features associated with specific brain regions and their corresponding functional characteristics. Formally, given an input

neural token $\mathbf{e}_{t,c}^l$ at time stamp $t$ and channel $c$ of the layer $l$, the computation proceeds as follows:

$$h_{t,c}^l = \text{MSA}\left(\text{Norm}\left(\mathbf{e}_{t,c}^l\right)\right) + \mathbf{e}_{t,c}^l, \tag{2}$$

$$\bar{h}_{t,c}^l = \text{Norm}\left(h_{t,c}^l\right), \tag{3}$$

$$\mathbf{e}_{t,c}^{l+1} = \left(\sum_{x=1}^{N_x} \Psi(\bar{h}_{t,c}^l) \cdot \text{FFN}_x(\bar{h}_{t,c}^l)\right) + h_{t,c}^l, \tag{4}$$

where MSA denotes the multi-head self-attention operation, $\text{Norm}$ is a normalization layer, and $\Psi(\cdot)$ represents the routing function indicating the contribution of each expert $\text{FFN}_x$.

To ensure that neural tokens originating from the same channel are consistently processed by a single expert, the routing function $\Psi(\cdot)$ operates in a channel-wise manner:

$$\hat{h}_{t \cdot c}^l = \sum_{t=1}^{\frac{T}{k}} \bar{h}_{t,c}^l, \tag{5}$$

$$\mathcal{D}_c = \text{Softmax}(\hat{h}_{t \cdot c}^l \cdot \Phi), \tag{6}$$

$$\Psi(\bar{h}_{t,c}^l) = \text{TopK}(\mathcal{D}_c, k), \tag{7}$$

where $\Phi \in \mathbb{R}^{d \times N_x}$ represents the trainable weight matrix of the router, and $\text{TopK}(\cdot, k)$ selects the top $k$ largest values, setting all others to zero. Unlike conventional MoE modules commonly employed in large language models, where the dispatch matrix $\mathcal{D}$ is computed based solely on individual tokens, Neuro-MoBRE aggregates neural feature information across the entire channel before routing, ensuring coherent channel-specific expert specialization. An auxiliary loss is adopted to ensure the load balance among experts (Lepikhin et al., 2020; Fedus et al., 2022; Shazeer et al., 2017).

**Task-disentangled Information Aggregation.** In Transformer-based architectures, a global learnable CLS token is typically employed to aggregate information from other tokens, subsequently processed by a feed-forward network (FFN) for further feature extraction. However, complex neural activities in humans involve distinct brain regions operating collaboratively under diverse functional patterns. Consequently, different neural decoding tasks require tailored neural feature aggregation strategies. To address this, we propose utilizing task-disentangled neural CLS tokens (denoted as $\text{CLS}_{\mathcal{T}_j}$) for each decoding task $\mathcal{T}_j$, each having a dimensionality $J$ times larger than standard local neural tokens. This approach enhances global neural representation capacity and facilitates more effective neural feature aggregation across brain regions (Fuller et al., 2025; Darcet et al., 2024). Formally, given an input neural token sequence $E \in \mathbb{R}^{\lfloor \frac{T}{k} \rfloor C \times d}$, we initialize $n$ distinct task-specific CLS tokens, each with dimensions $J \cdot d$. Prior to applying the multi-head self-attention (MSA) operation, each neural CLS token is decomposed into $J$ separate $d$-dimensional tokens. These are concatenated with the existing local neural tokens, forming the expanded token sequence $E^{\text{CLS}} \in \mathbb{R}^{(\lfloor \frac{T}{k} \rfloor C + nJ) \times d}$. Following the MSA operation, neural CLS tokens are separated from the combined sequence to yield:

$$E^{\text{CLS}} \to \left(\text{CLS}_{\mathcal{T}_1} \in \mathbb{R}^{Jd}, ..., \text{CLS}_{\mathcal{T}_n} \in \mathbb{R}^{Jd}, \mathbb{R}^{\lfloor \frac{T}{k} \rfloor C \times d}\right). \tag{8}$$

Given the fundamental difference between extracting neural patterns from individual brain regions and aggregating neural features for distinct decoding tasks, the neural CLS tokens are processed using a specialized FFN. Conversely, local neural tokens are routed through the $\text{FFN}s$ contained within the Brain-regional MoE module. Finally, we employ lightweight task-specific neural decoders that map the processed CLS tokens from the final layer to the corresponding decoding class labels.

## 3.2 Regional Masked Autoencoding as Co-upcycling Initialization

Thus far, we have managed to address the neural heterogeneity from diverse electrode implantation strategies and variations in neural patterns corresponding to distinct neural activities. To further mitigate inter-subject neural variability and establish a unified, generalizable neural feature extractor, we propose a regional masked autoencoding (RMAE) approach, shown in Fig. 2. Specifically, we first pre-train a set of $m$ subject-specific models parameterized by $\{\theta_i\}_{i=1}^m$ via the proposed RMAE task. These models share the original network structure described earlier, except that the brain-regional mixture-of-experts (MoE) block is replaced by a conventional FFN. Subsequently, we "co-upcycle"

these individual subject-specific pre-trained models as initialization for the proposed Neuro-MoBRE framework, enhancing its cross-subject generalization capability. Given a neural token sequence $E$, we randomly mask all tokens of the same brain region with a masking probability ratio $r$. During pre-training, the masked tokens are replaced by learnable embedding tokens of the same feature dimension $d$. A lightweight prediction head is then employed to predict the frequency-domain representation, with the training objective defined as the mean squared error between the original and predicted time-domain signals of the masked regions (Wu et al., 2024). We set $r = 0.2$ by default in this work. We provide a detailed calculation of the prediction target in the Appendix D. We then apply "co-upcycle" to the neural feature extraction network components while randomly initializing the brain-region-specific experts to increase their diversity and degrees of freedom, enhancing representational distinctions across different brain regions and tasks during feature extraction.

For Co-upcycle, we consider two variants: naïve averaging and sign-uniformity aggregation (Yadav et al., 2023). Naïve averaging computes the element-wise mean of all subject-specific models, given by $\sum_{i=1}^{m} \theta_i / m$. In contrast, sign-uniformity co-upcycle first sparsifies each pre-trained model by pruning the lowest-magnitude parameters, setting the smallest $k\%$ to zero, yielding trimmed parameters $\{\hat{\theta}_i\}_{i=1}^{m}$. For each parameter $p$, we then establish a consensus sign $\gamma^p = \text{sgn}(\sum_{i=1}^{m} \hat{\theta}_i^p)$, where sgn denotes the sign function. Defining the index set $\mathcal{I}^p = \{i \mid \text{sgn}(\hat{\theta}_i^p) = \gamma^p\}$, we initialize the parameter $p$ in the "co-upcycled" network as $\theta_{\text{init}}^p = \frac{1}{|\mathcal{I}^p|} \sum_{i \in \mathcal{I}^p} \hat{\theta}_i^p$. The proposed initialization algorithm does not incur any extra computational overhead. A detailed complexity analysis is presented in the Appendix E.

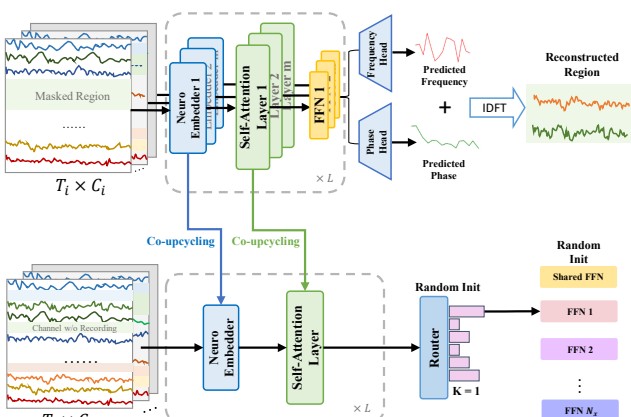

Figure 2: Illustration of the RMAE initialization strategy. Subject-specific neuro-embeddings and attention layers pre-trained via RMAE, utilizing frequency and phase distributions as prediction objectives, are jointly leveraged to initialize the Neuro-MoBRE model parameters. The shared FFN and BrMoE FFN remain randomly initialized to enable effective feature specialization during fine-tuning.

## 4 EXPERIMENTS

### 4.1 EXPERIMENTAL SETUP

**Decoding tasks and data acquisition.** In this study, we focus on neural decoding tasks relevant to language decoding and epileptic seizure diagnosis. We recruited 11 participants who were undergoing epilepsy monitoring with implanted stereo-electroencephalography (sEEG) electrodes at anonymous hospitals (hospital names omitted to maintain compatibility with double-blind review). Figure A1 illustrates the electrode distributions across subjects, while Table A1 summarizes participant demographic information. Experimental protocols were approved by institutional review boards, and participants provided written informed consent. 1) **Language decoding task**: Participants read aloud 407 monosyllabic Mandarin characters repeated three times, capturing common Mandarin pronunciations. Mandarin syllables contain three phonological components: initials (23 classes), finals (11 classes), and tones (4 classes), each decoded as separate classification tasks. Given the acoustic similarity and complexity among Mandarin's 35 phonetic finals, we grouped them into 11 clusters based on acoustic features via k-means clustering Feng et al. (2023), simplifying decoding. For analyses, we excluded the electrodes located in the visual cortex or white matter. 2) **Epileptic seizure diagnosis tasks**: In the epilepsy diagnosis tasks, we addressed both detection and prediction. Seizure detection is formulated as a classification task of differentiating ictal (seizure-active) from interictal (seizure-free) brain states. Seizure prediction aims to identify electrophysiological patterns or biomarkers indicative of imminent seizure onset, allowing for estimation of seizure likelihood

Table 1: Performance comparison for language decoding tasks. Multi-sub denotes multi-subject. The best and second-best results are highlighted in **bold** and underlined, respectively. "Avg" and "Sgn" to the averaging and sign-uniformity variants of Co-upcycle, respectively.

| Methods | Multi-sub | Multi-task | Initial Decoding | | Finals Decoding | | Tone Decoding | |
|---|---|---|---|---|---|---|---|---|
| | | | Accuracy | Cohen's Kappa | Accuracy | Cohen's Kappa | Accuracy | Cohen's Kappa |
| ST-Transformer (Song et al., 2021) | ✗ | ✗ | $0.1178 \pm 0.0639$ | $0.0777 \pm 0.0668$ | $0.1434 \pm 0.0819$ | $0.0983 \pm 0.0862$ | $0.2864 \pm 0.0910$ | $0.2792 \pm 0.0919$ |
| FFCL (Li et al., 2022) | ✗ | ✗ | $0.1055 \pm 0.0711$ | $0.0649 \pm 0.0744$ | $0.1277 \pm 0.0748$ | $0.0818 \pm 0.0787$ | $0.2857 \pm 0.0884$ | $0.2785 \pm 0.0893$ |
| CoST (Woo et al., 2022) | ✗ | ✗ | $0.1443 \pm 0.0796$ | $0.1054 \pm 0.0832$ | $0.1583 \pm 0.0984$ | $0.1140 \pm 0.1035$ | $0.2973 \pm 0.0736$ | $0.2902 \pm 0.0743$ |
| SPaRCNet (Jing et al., 2023) | ✗ | ✗ | $0.0921 \pm 0.0761$ | $0.0508 \pm 0.0796$ | $0.1087 \pm 0.0871$ | $0.0618 \pm 0.0917$ | $0.2609 \pm 0.0826$ | $0.2534 \pm 0.0834$ |
| BIOT (Yang et al., 2023) | ✓ | ✗ | $0.1478 \pm 0.0683$ | $0.1091 \pm 0.0715$ | $0.1736 \pm 0.0899$ | $0.1301 \pm 0.0946$ | $0.3209 \pm 0.0817$ | $0.3140 \pm 0.0825$ |
| Brant (Zhang et al., 2023) | ✓ | ✗ | $0.1423 \pm 0.0923$ | $0.1033 \pm 0.0964$ | $0.1963 \pm 0.0699$ | $0.1540 \pm 0.0736$ | $0.3548 \pm 0.0837$ | $0.3483 \pm 0.0845$ |
| NeuroBERT (Wu et al., 2024) | ✗ | ✗ | $0.1312 \pm 0.0620$ | $0.0917 \pm 0.0648$ | $0.2029 \pm 0.0936$ | $0.1609 \pm 0.0985$ | $0.3223 \pm 0.08851$ | $0.3154 \pm 0.0894$ |
| LaBraM (Jiang et al., 2024) | ✓ | ✗ | $0.1300 \pm 0.0660$ | $0.0905 \pm 0.0690$ | $0.1942 \pm 0.0754$ | $0.1518 \pm 0.0794$ | $0.3334 \pm 0.0873$ | $0.3267 \pm 0.0882$ |
| H2DiLR (Wu et al., 2025) | ✗ | ✗ | $0.1561 \pm 0.1229$ | $0.1178 \pm 0.1285$ | $0.2169 \pm 0.0696$ | $0.1757 \pm 0.0733$ | $0.3277 \pm 0.0843$ | $0.3209 \pm 0.0852$ |
| PopT (Chau et al., 2025) | ✓ | ✗ | $0.0818 \pm 0.0751$ | $0.0335 \pm 0.0790$ | $0.1686 \pm 0.0931$ | $0.1248 \pm 0.0980$ | $0.2632 \pm 0.0847$ | $0.2557 \pm 0.0856$ |
| NeuroLM (Jiang et al., 2025) | ✓ | ✓ | $0.0621 \pm 0.0826$ | $0.0194 \pm 0.0864$ | $0.1409 \pm 0.0889$ | $0.0957 \pm 0.0936$ | $0.2766 \pm 0.0873$ | $0.2693 \pm 0.0882$ |
| Neuro-MoBRE (Avg Co-upcycle) | ✓ | ✓ | $\underline{0.2810 \pm 0.0898}$ | $\underline{0.2011 \pm 0.0998}$ | $\underline{0.2860 \pm 0.0973}$ | $\underline{0.2484 \pm 0.1024}$ | $\underline{0.4125 \pm 0.0926}$ | $\underline{0.4051 \pm 0.0938}$ |
| Neuro-MoBRE (Sgn Co-upcycle) | ✓ | ✓ | $\mathbf{0.2901 \pm 0.0933}$ | $\mathbf{0.2112 \pm 0.1037}$ | $\mathbf{0.3161 \pm 0.0901}$ | $\mathbf{0.2801 \pm 0.0949}$ | $\mathbf{0.4341 \pm 0.0813}$ | $\mathbf{0.4284 \pm 0.0821}$ |

Table 2: Performance comparison for seizure prediction and detection tasks. The best and second-best results are highlighted in **bold** and underlined, respectively. "Avg" and "Sgn" to the averaging and sign-uniformity variants of Co-upcycle, respectively.

| Methods | Multi-sub | Multi-task | Seizure Prediction | | | Seizure Detection | | |
|---|---|---|---|---|---|---|---|---|
| | | | Accuracy | Sensitivity | Weighted F1 | Accuracy | Sensitivity | Weighted F1 |
| ST-Transformer (Song et al., 2021) | ✗ | ✗ | $0.6357 \pm 0.1002$ | $0.6414 \pm 0.1077$ | $0.6356 \pm 0.1001$ | $0.7170 \pm 0.1003$ | $0.7127 \pm 0.1051$ | $0.7170 \pm 0.1002$ |
| FFCL (Li et al., 2022) | ✗ | ✗ | $0.6295 \pm 0.1154$ | $0.6355 \pm 0.1120$ | $0.6293 \pm 0.1157$ | $0.6902 \pm 0.1121$ | $0.6900 \pm 0.1028$ | $0.6901 \pm 0.1122$ |
| CoST (Woo et al., 2022) | ✗ | ✗ | $0.6734 \pm 0.1022$ | $0.6795 \pm 0.1116$ | $0.6731 \pm 0.1024$ | $0.7482 \pm 0.1122$ | $0.7491 \pm 0.1178$ | $0.7480 \pm 0.1123$ |
| SPaRCNet (Jing et al., 2023) | ✗ | ✗ | $0.6361 \pm 0.1097$ | $0.6255 \pm 0.1129$ | $0.6358 \pm 0.1097$ | $0.6845 \pm 0.0982$ | $0.6745 \pm 0.1014$ | $0.6843 \pm 0.0984$ |
| BIOT (Yang et al., 2023) | ✓ | ✗ | $0.6595 \pm 0.1139$ | $0.6627 \pm 0.1161$ | $0.6594 \pm 0.1140$ | $0.8011 \pm 0.1078$ | $0.7959 \pm 0.1107$ | $0.8011 \pm 0.1078$ |
| Brant (Zhang et al., 2023) | ✓ | ✗ | $0.6602 \pm 0.1143$ | $0.6695 \pm 0.1108$ | $0.6600 \pm 0.1146$ | $0.7850 \pm 0.1000$ | $0.7945 \pm 0.0963$ | $0.7848 \pm 0.1001$ |
| NeuroBERT (Wu et al., 2024) | ✗ | ✗ | $0.6518 \pm 0.1024$ | $0.6536 \pm 0.1123$ | $0.6517 \pm 0.1025$ | $0.7614 \pm 0.1046$ | $0.7550 \pm 0.0994$ | $0.7612 \pm 0.1046$ |
| LaBraM (Jiang et al., 2024) | ✓ | ✗ | $0.6775 \pm 0.1101$ | $0.6795 \pm 0.1169$ | $0.6773 \pm 0.1102$ | $0.7989 \pm 0.1130$ | $0.7991 \pm 0.1076$ | $0.7988 \pm 0.1131$ |
| H2DiLR (Wu et al., 2025) | ✗ | ✗ | $0.6734 \pm 0.1008$ | $0.6827 \pm 0.1041$ | $0.6733 \pm 0.1008$ | $0.7682 \pm 0.1044$ | $0.7655 \pm 0.1126$ | $0.7681 \pm 0.1045$ |
| PopT (Chau et al., 2025) | ✓ | ✗ | $0.6614 \pm 0.1150$ | $0.6695 \pm 0.1146$ | $0.6612 \pm 0.1149$ | $0.7652 \pm 0.0887$ | $0.7600 \pm 0.0908$ | $0.7652 \pm 0.0887$ |
| NeuroLM (Jiang et al., 2025) | ✓ | ✓ | $0.6495 \pm 0.1057$ | $0.6364 \pm 0.1166$ | $0.6492 \pm 0.1060$ | $0.6927 \pm 0.1192$ | $0.6864 \pm 0.1319$ | $0.6925 \pm 0.1193$ |
| Neuro-MoBRE (Avg co-upcycle) | ✓ | ✓ | $\underline{0.7636 \pm 0.1068}$ | $\underline{0.7655 \pm 0.1140}$ | $\underline{0.7636 \pm 0.1068}$ | $\underline{0.8730 \pm 0.1069}$ | $\underline{0.8627 \pm 0.1117}$ | $\underline{0.8729 \pm 0.1070}$ |
| Neuro-MoBRE (Sgn co-upcycle) | ✓ | ✓ | $\mathbf{0.7877 \pm 0.1044}$ | $\mathbf{0.7741 \pm 0.1092}$ | $\mathbf{0.7877 \pm 0.1044}$ | $\mathbf{0.8957 \pm 0.0900}$ | $\mathbf{0.8991 \pm 0.0882}$ | $\mathbf{0.8957 \pm 0.0900}$ |

within a predefined future time window. We set a seizure prediction horizon (SPH) at 5 minutes, a preictal interval length (PIL) at 35 minutes Wu et al. (2022b), and defined interictal intervals from 30 minutes after seizure offset to the next preictal interval. Clinical experts annotated seizure onset and offset via visual inspection of sEEG signals. Interictal samples were generated from non-overlapping 20-second sliding windows. For preictal and ictal samples, we employed overlapping windows (50% overlap) to address sample imbalance. We provide detailed signal preprocessing in Appendix C.

**Evaluation Protocols.** For the language decoding tasks (formulated as multi-class classification problems), decoding performance was evaluated using top-1 accuracy (Acc) and Cohen's Kappa coefficient. For the epilepsy diagnosis tasks, we assessed top-1 accuracy (Acc), sensitivity (Sen), and weighted F1-score. Regarding dataset partitioning, we employed a balanced sampling strategy to ensure uniform representation across classes within the test set. Specifically, for language decoding tasks, we randomly sampled 10, 20, and 80 instances per class from each participant to construct the test datasets for initial decoding, final decoding, and tone decoding, respectively. This sampling approach resulted in test sets comprising approximately 20% of the total available data. The remaining samples formed the training datasets, from which we further allocated 20% to create the validation sets. For epileptic seizure diagnosis tasks, participant data were partitioned into training (80%), and testing (20%) subsets. From the training subset, an additional 20% was reserved as a validation set. The sliding-window segmentation approach with overlapping windows applied during preprocessing inherently addressed potential class imbalance. All experiments were conducted five times using distinct random seeds. We report performance results as mean values with corresponding standard deviations across these repetitions. We provide implementation details in Appendix H.

### 4.2 DECODING PERFORMANCE COMPARISON

We compare the performance of our proposed Neuro-MoBRE with baseline methods in Tab. 1 and 2. In language decoding tasks, conventional supervised pre-training baselines achieve decoding performance slightly above chance levels. Baselines employing pre-training consistently outperform those

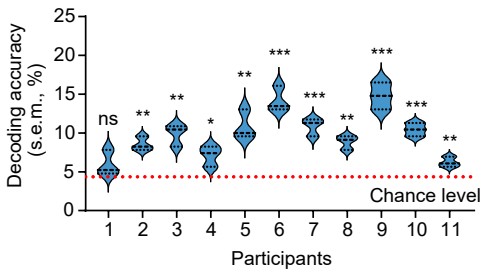

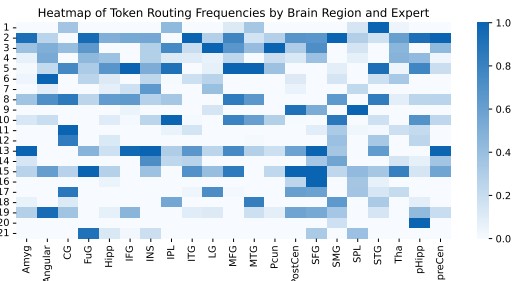

Figure 3: Initial decoding performance of Neuro-MoBRE on unseen subjects.

Figure 4: Distribution of brain regions processed by different experts.

without pre-training, primarily due to enhanced generic feature extraction capabilities. Models such as BIOT, LaBraM, and H2DiLR exhibit superior decoding performance, benefiting from sophisticated architectures designed at both micro and macro levels. However, most existing methods assume fixed electrode placements across subjects to facilitate cross-subject decoding. This assumption limits their practical application in intracranial implant scenarios, where electrode configurations vary significantly between subjects. Consequently, these approaches are marked as incapable of multi-subject decoding in realistic scenarios. Originally developed for tasks like abnormal detection Harati et al. (2015) and sleep stage classification Alvarez-Estevez & Rijsman (2021), which feature relatively consistent neural patterns (e.g., specific frequency activations), these methods struggle with tasks such as language decoding that involve higher inter-subject neural variability. Although frameworks like BIOT, LaBraM, and NeuroLM address data heterogeneity related to differences in channel counts, sampling rates, and data lengths, they fall short in handling variability stemming from distinct neural activity patterns and substantial inter-subject differences. Despite NeuroLM establishing a multi-task decoding paradigm using a question-answer interaction, it still inadequately captures subtle neural pattern differences, thereby limiting its effectiveness due to unresolved neural data heterogeneity. Our proposed Neuro-MoBRE framework explicitly addresses these sources of neural data heterogeneity throughout its design, leading to significantly improved decoding performance compared to existing approaches. Notably, performance variability in language decoding tasks largely arises from subject-specific differences in phoneme articulation. For example, Participant 4's lower decoding accuracy ($\sim$20%) is primarily due to dialect-influenced articulation challenges, specifically difficulty distinguishing phonemes such as 's' and 'sh'. In contrast, Participant 10 demonstrates markedly higher decoding accuracy ($\sim$60%), reflecting lower phoneme articulation variability.

### 4.3 ANALYSIS ON NEURO-MOBRE

**Neuro-MoBRE generalizes to unseen subject.** A key advantage of multi-subject, multi-task decoding models over conventional single-task approaches is their improved ability to learn subject-invariant representations and effectively handle inter-subject variability. To systematically evaluate Neuro-MoBRE in terms of generalization to unseen subjects, we conducted experiments following a leave-one-subject-out (LOSO) validation scheme. Specifically, we trained Neuro-MoBRE exclusively on neural data from a subset of subjects and subsequently assessed performance on subjects whose data were completely withheld from training. Our experimental protocol closely follows the implementation details outlined earlier, and we present the decoding performance results for this LOSO evaluation in Fig. 3. Our results demonstrate that Neuro-MoBRE consistently achieves decoding performance significantly above chance-level on previously unseen subjects, underscoring its effectiveness in learning robust, subject-agnostic neural features. Crucially, by successfully generalizing across subject-specific variability, Neuro-MoBRE highlights its potential as a powerful tool for reliable, real-world deployment across diverse, unseen populations in neural decoding tasks.

**Regional experts focus on specialized brain regions.** To examine whether regional experts within our proposed model exhibit clear preferences toward particular brain regions, we analyzed and visualized the distributions of tokens from these regions as processed by each expert. As illustrated in Fig. 4, the token distribution significantly varies across different experts, demonstrating specialized functional roles. A darker color for a given brain region–expert pair indicates a higher routing intensity, signifying that more tokens from the corresponding brain region are routed to that expert. For instance, Expert One predominantly focuses on processing neural signals from the superior temporal gyrus

Table 3: Ablation study on different components of Neuro-MoBRE.

| Component | Decoding Task | | |
| --- | --- | --- | --- |
| | Initial Decoding | Final Decoding | Tone Decoding |
| Tokenizer | 0.1798 | 0.2256 | 0.3270 |
| + BrMoE | 0.2483 | 0.2798 | 0.3639 |
| + TIA | 0.2719 | 0.3025 | 0.4045 |
| + RMAE | 0.2901 | 0.3161 | 0.4341 |

Table 4: Ablation study on different number of experts.

| Number of Experts | Decoding Task | | |
| --- | --- | --- | --- |
| | Initial Decoding | Final Decoding | Tone Decoding |
| 4 | 0.2336 | 0.2696 | 0.3736 |
| 8 | 0.2526 | 0.2905 | 0.3955 |
| 16 | 0.2751 | 0.3112 | 0.4185 |
| 21 | 0.2901 | 0.3161 | 0.4341 |

(STG) related to auditory processing related to speech and language, while Experts 2, 8, 13, and 15 are more involved with pre-central and post-central gyrus associated with speech production, assisting motor areas in fine-tuning and coordinating speech movements. Such selective processing further validates that regional experts function by attending to specialized brain areas, aligning effectively with the functional modularity observed in neurobiological studies. This specialization enables accurate extraction of region-specific features and facilitates targeted decoding of neural activities.

### 4.4 ABLATION STUDY

This section ablates three key designs and the effect of different numbers of experts on the task of language decoding. The average top-1 accuracy for initial decoding is reported, and we follow the same experimental setup as described in Appendix H. Additional ablation study on the scaling effect of network parameters is provided in Appendix I.1.

**Effectiveness of the proposed components.** Next, we conduct an ablation study to verify the effectiveness of different components proposed in Neuro-MoBRE. The brain-regional-temporal tokenizer is fundamental to accommodating varied electrode implantation schemes and generating unified neural embeddings; thus, we treat it as non-removable and do not ablate this component. Specifically, we assess the contributions of the brain-regional Mixture-of-Experts block (BrMoE), the task-disentangled information aggregation mechanism (TIA), and the regional masked autoencoding strategy (RMAE). As illustrated in Tab. 3, both BrMoE and TIA substantially improve the decoding performance. Moreover, incorporating the proposed RMAE strategy further enhances the decoding results. These findings empirically support our hypothesis that effectively addressing data heterogeneity is essential for multi-subject and multi-task neural decoding.

**Ablation on the number of experts.** Ideally, neural features from distinct brain regions should be modeled by dedicated experts. We investigate how varying the number of experts impacts overall decoding performance. As observed in Tab. 4, increasing the number of experts generally leads to improved performance, whereas using fewer experts results in performance degradation. Interestingly, comparable decoding accuracies were obtained from 8 and 16 experts. We hypothesize that, with fewer experts than brain regions, Neuro-MoBRE captures collaborative patterns among brain regions that function jointly as coherent groups during specific neural activities. Conversely, when the number of experts exceeds the number of brain regions, multiple experts could be dedicated adaptively to brain regions that contribute more significantly to underlying neural processes.

### 5 CONCLUSION AND LIMITATION

This paper presents the Neural Mixture of Brain Regional Experts (Neuro-MoBRE), a novel framework designed to explicitly resolve the ubiquitous data heterogeneity encountered in multi-task and multi-subject neurophysiological decoding. Extensive experimental evaluations conducted on stereo-electroencephalography (sEEG) data collected from a cohort of 11 subjects and spanning five distinct decoding tasks, including language decoding and seizure diagnosis, showed that Neuro-MoBRE substantially outperforms existing decoding methods. We list three potential limitations of this work: (1) We have currently verified the effectiveness of Neuro-MoBRE on language decoding and epileptic seizure diagnosis; extending it to more diverse tasks that capture broader neural behaviors could facilitate a deeper understanding of general brain functionality and neural representations. (2) Currently, Neuro-MoBRE treats the neural patterns of an entire brain region as the smallest modeling unit; however, more detailed functional modeling at the level of smaller anatomical areas within each region has yet to be investigated. (3) The inherent complexity and invasiveness involved in acquiring intracranial recordings constrain our capacity to expand the participant cohort at this time.

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

## A DETAILED RELATED WORK

**Intracranial Neurophysiological Decoding.** Intracranial brain recordings provide an unparalleled view into the brain's electrical activity, enabling detailed analysis of complex neural processes inaccessible to non-invasive methods such as EEG. These recordings typically involve electrodes placed directly on the brain's surface (electrocorticography, ECoG) or inserted into deeper brain structures (stereo-electroencephalography, sEEG). Unlike scalp EEG, which is significantly attenuated by the skull and scalp, intracranial recordings offer superior spatial and temporal resolution. This makes them particularly suited for pinpointing localized neural activity and investigating the neural substrates of cognitive, sensory, and motor functions (Bouton et al., 2016; Benabid et al., 2019; Hochberg et al., 2012; Willett et al., 2023; Metzger et al., 2023). The investigation of intracranial brain recordings began as early as 1874 when Roberts Bartholow first documented the effects of electrical stimulation on human brain tissue and its subsequent impact on motor functions (Harris & Almerigi, 2009). More recently, Hochberg et al. (Hochberg et al., 2012) advanced this field by developing a neural interface that enabled two individuals with long-standing tetraplegia to control robotic arms using signals collected from a 96-channel microelectrode array implanted in the motor cortex. This groundbreaking work demonstrated participants' ability to execute complex tasks, including three-dimensional reach-and-grasp movements, with one participant even successfully drinking from a bottle. Beyond motor restoration, intracranial brain recordings have been instrumental in developing speech prostheses aimed at restoring rapid communication for individuals experiencing paralysis (Willett et al., 2023; Metzger et al., 2023; Moses et al., 2021; Angrick et al., 2019). By decoding neural activity associated with attempted speech, these systems convert neural signals into text or audible speech. Notably, Willett et al. introduced a speech prosthesis utilizing intracranial recordings from microelectrode arrays implanted in the ventral premotor cortex (area 6v) and Broca's area (area 44). Their system achieved impressive word error rates, 9.1% on a 50-word vocabulary and 23.8% on a 125,000-word vocabulary, representing substantial progress in speech decoding capabilities. Despite these notable advances, most studies have predominantly focused on decoding neural signals from individual subjects engaged in single-task scenarios, limiting the generalizability and broader applicability of these methods. To address this limitation, we propose a unified neurophysiological decoding framework capable of multi-task decoding by integrating intracranial recordings collected from a diverse cohort of 11 subjects, thereby enhancing the model's versatility and robustness across various neural decoding tasks.

**Large-scale Neurophysiological Model.** Large language models (LLMs) have demonstrated remarkable abilities as general-purpose task solvers when trained on extensive corpora (Brown et al., 2020). Motivated by these achievements, several researchers have explored methods for learning robust neurophysiological representations by integrating EEG signals collected from multiple subjects across diverse decoding tasks. For instance, MMM (Yi et al., 2023) introduced a topology-agnostic pre-training framework, remapping EEG recordings from various sensor configurations into a standardized 10-10 system topology through multi-dimensional positional encodings, effectively incorporating geometric information. MMM employs a masked autoencoding paradigm at both channel and regional levels as its pretext task. Similarly, BIOT (Yang et al., 2023) proposed a handcrafted tokenizer capable of converting biosignals of arbitrary lengths and channel arrangements into a unified neural representation across subjects, also leveraging channel-wise and temporal-wise masking within a masked autoencoding framework. Furthermore, both Brant (Zhang et al., 2024) and BrainBERT (Wang et al., 2023) adopt masked autoencoding strategies in the frequency domain, utilizing intracranial recordings as inputs. LaBraM (Jiang et al., 2024) introduced discrete tokenization of continuous EEG signals to serve as prediction targets within their masked autoencoding framework. Despite leveraging neurophysiological signals from multiple subjects across varied decoding scenarios, these approaches require extensive fine-tuning for each downstream task and generally lack the capability for effective multi-task decoding. NeuroLM (Jiang et al., 2025) addressed this limitation by reframing EEG decoding as a question-answering problem facilitated by large language models, demonstrating multi-task capabilities. Nevertheless, existing approaches predominantly focus on designing encoder architectures to reconcile differences in data format but often neglect the explicit modeling of data heterogeneity arising from variations between subjects and tasks. Recently, H2DiLR (Wu et al., 2025) addressed heterogeneity explicitly by disentangling homogeneous and heterogeneous components in EEG signals, enabling unified decoding across subjects but restricted to single-task decoding scenarios. In this paper, we introduce a novel multi-task, multi-subject neurophysiological decoding framework equipped with explicit disentanglement mechanisms to

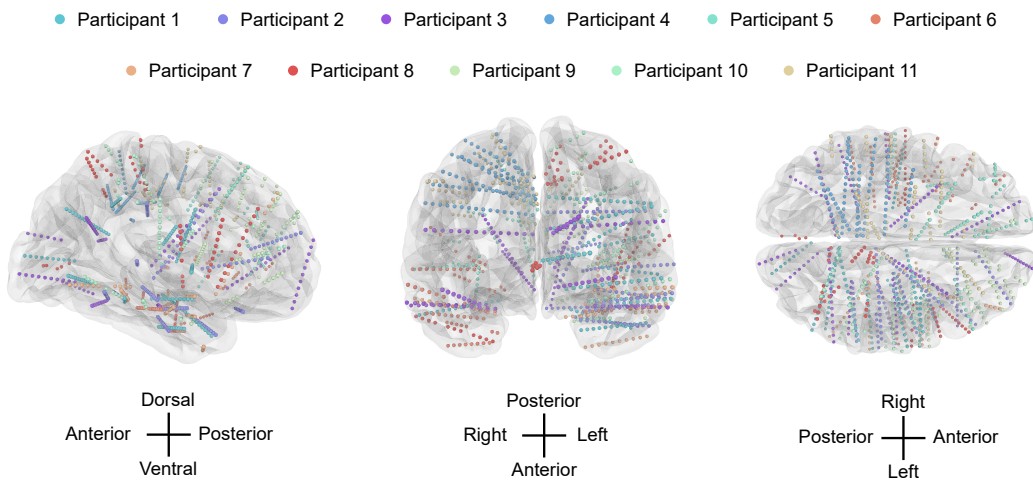

Figure A1: The anatomy of 11 participants mapped onto the standard Montreal Neurological Institute template brain, with directions indicated.

Table A1: Demographic and experimental information of participants, comprising sex, age, handedness, and the number of implanted electrodes.

| Participant | Sex | Age | Handedness | Number of electrodes | Channel count |
|---|---|---|---|---|---|
| 1 | Male | 20 | Right-handed | 12 | 63 |
| 2 | Male | 25 | Right-handed | 8 | 36 |
| 3 | Female | 19 | Right-handed | 11 | 30 |
| 4 | Female | 19 | Right-handed | 13 | 66 |
| 5 | Male | 14 | Right-handed | 9 | 73 |
| 6 | Female | 17 | Left-handed | 11 | 78 |
| 7 | Female | 15 | Right-handed | 11 | 74 |
| 8 | Male | 15 | Right-handed | 12 | 68 |
| 9 | Female | 26 | Right-handed | 12 | 61 |
| 10 | Male | 35 | Right-handed | 12 | 88 |
| 11 | Male | 22 | Right-handed | 10 | 84 |

address data heterogeneity stemming from both task-specific and subject-specific factors, enhancing decoding versatility and generalizability.

## B  DATA ACQUISITION ON LANGUAGE DECODING TASK

Participants were instructed to read aloud a set of 407 monosyllabic Mandarin characters, repeated three times. These characters were selected to comprehensively cover the range of common pronunciations in Mandarin Chinese. Each Mandarin syllable comprises three distinct phonological elements: an initial consonant, a final vowel (or vowel combination), and a lexical tone. Accordingly, we defined the decoding procedure in terms of three classification tasks: initials (23 classes), finals (11 classes), and tones (4 classes). Given that Mandarin contains 35 final phonemes (6 simple finals and 29 compound finals), the high acoustic similarity and dynamic articulation involved rendered direct classification challenging. To overcome this issue, we employed k-means clustering on acoustic formant-based features derived from the original 35 final phonemes Feng et al. (2023). This process grouped acoustically and articulatorily similar finals, resulting in 11 distinct final clusters conducive to improved decoding performance. For all language decoding analyses, we specifically selected channels associated with speech production regions and excluded electrodes located in the visual

cortex or within the white matter. During recording, the participants read aloud the target character displayed on a screen following a countdown visual cue. Each reading trial consisted of a preparation phase lasting 0.8–1.2 seconds, followed by an articulation phase of 2 seconds, and concluded with a rest period of 0.5–1.0 seconds. A comprehensive list of characters along with their corresponding syllables is presented in Tab. A5–A8. These characters were selected to comprehensively cover the range of common pronunciations in Mandarin Chinese. Our reading material has been carefully designed by Mandarin linguists to encompass as many pronunciation phenomena as possible, thereby ensuring that the brain decoding algorithm achieves broad generalizability.

## C  NEURAL RECORDING PREPROCESSING.

The neural recordings were first converted to microvolts ($\mu V$) to ensure consistency in magnitude. The raw data were then filtered using a low-pass filter with a cutoff frequency of 200 Hz to prevent aliasing artifacts. Additionally, notch filters centered at 50 Hz and its harmonics at 100 Hz were applied to remove power-line interference. Following these filtering procedures, the neural signals were downsampled to 512 Hz to reduce computational overhead. Finally, each channel's data was individually normalized by subtracting its mean and dividing by its standard deviation, standardizing the signals to facilitate subsequent analyses.

## D  PREDICTION TARGET OF RMAE.

Given a neural token sequence $E$, we randomly select and mask tokens that belong to the same brain region with a masking ratio $r$ sampled uniformly. During pre-training, the masked tokens are replaced by learnable embedding tokens of the same feature dimension $d$. Suppose that $x$ is the corresponding discrete neurological signal of the masked patch, we apply the discrete Fourier transform (DFT) as:

$$X_m = \sum_{n=1}^{n=N} x_n * e^{-\frac{2\pi j}{N}mn}, \tag{9}$$

where $m \in [1, N]$. We further decompose equation 9 into its real and imaginary components by applying Euler's formula:

$$X_m = \sum_{n=1}^{n=N} x_n * \underbrace{\cos(\frac{2\pi}{N}mn)}_{\text{real}} - \underbrace{j * \sin(\frac{2\pi}{N}mn)}_{\text{imaginary}}, \tag{10}$$

where $j$ denotes the imaginary unit, defined by $j^2 = -1$. Specifically, $X_m$ represents the spectrum of the sequence $x_n$ evaluated at frequency $\omega_m = 2\pi m/N$. Consequently, we can compute the magnitude $\|X_m\|^2$ and phase $\theta_m$ as follows:

$$\|X_m\|^2 = \frac{1}{N}\sqrt{\text{Re}(X_m)^2 + \text{Im}(X_m)^2}$$
$$\theta_m = \text{atan2}(\text{Re}(X_m)^2, \text{Im}(X_m)^2), \tag{11}$$

where $\text{atan2}(\cdot, \cdot)$ denotes the two-argument arctangent function, and $\text{Re}(\cdot)$ and $\text{Im}(\cdot)$ represent the real and imaginary parts, respectively. We define our prediction target as the missing amplitude and phase components of the signal. Subsequently, we apply the inverse discrete Fourier transform (IDFT) to convert the predicted amplitude and phase back into the temporal domain:

$$\hat{X}_m = \left\|\hat{X}_m\right\|^2 * e^{j*\hat{\theta}_m}$$
$$\hat{x}_n = \sum_{m=1}^{m=N} \hat{X}_m * e^{\frac{2\pi j}{N}mn}. \tag{12}$$

A lightweight prediction head is then employed to predict the frequency-domain representation, with the training objective defined as the mean squared error between the original and predicted time-domain signals of the masked regions Wu et al. (2024). We set $r = 0.2$ by default in this work.

## E   TRAINING COMPLEXITY ANALYSIS OF CO-UPCYCLE INITIALIZATION

Please note that the proposed Co-upcycle mechanism is motivated by the hypothesis that corresponding brain regions across individuals exhibit functional homogeneity. Accordingly, Co-upcycle is applied only to the neural embedding layer and attention module parameters pretrained on different subjects via RMAE. In what follows, we analyze the computational and spatial complexity of Co-upcycle.

The overall RMAE Co-upcycle initialization can be divided into two parts: (i) the RMAE pretraining stage and (ii) the parameter aggregation (Co-upcycle) stage. We first examine the computational complexity of the RMAE component. Assume data are collected from $m$ subjects, each with $N$ data samples and let the model contain $P$ parameters. For simplicity, we assume that a single forward-backward training step incurs a constant factor number of FLOPs per parameter per sample. Under the conventional homogeneous pretraining setting, where data from all subjects are pooled and used to train one universal model, as adopted in prior work (Chau et al., 2025; Jiang et al., 2024; Zhang et al., 2023; Yang et al., 2023), the total training complexity is $O(mN \cdot P)$.

In our proposed Co-upcycle initialization, we instead pretrain $m$ subject-specific models, each with complexity $O(NP)$. The total pretraining cost is therefore also $O(m \cdot NP)$, assuming an equal number of epochs in both approaches. In practice, subject-specific models often converge in fewer steps than a single universal model, further reducing the effective training burden. For the Co-upcycle stage, the naïve averaging and sign-uniformity Co-upcycle require only $O(P \cdot m)$ FLOPs in total, which is negligible compared with the $O(m \cdot NP)$ cost of pretraining. Hence, Co-upcycle introduces no additional asymptotic training complexity beyond that of homogeneous pretraining.

Regarding spatial complexity, homogeneous pretraining requires $O(P)$ memory. The naïve averaging variant of Co-upcycle incurs the same spatial cost, as it only computes the element-wise mean across models. In contrast, the sign-uniformity variant in its naïve form requires $O(P \cdot m)$ storage, since it must hold all m subject-specific models simultaneously. However, in realistic scenarios, where data from all m subjects have already been collected, an efficient implementation can be adopted. By maintaining only running aggregates, namely the positive/negative sign counts and corresponding sums, one needs to store only four additional matrices of size $P$. Consequently, the spatial complexity reduces to $O(P)$, independent of the number of subjects $m$. Thus, Co-upcycle can be implemented with spatial cost asymptotically identical to homogeneous pretraining, i.e., $O(P)$. This demonstrates that scalability is not constrained by memory, as the framework avoids persistent storage of all subject-specific models.

## F   NETWORK ARCHITECTURE SETTINGS

We list the detailed description of network architectures in Tab. A2. We set $J$ and $K$ to 4 and 2 by default, respectively. Since most of the baseline methods are not inherently designed to handle data heterogeneity arising from varying channel counts across subjects, we introduce a convolutional layer specifically to project the differing channel dimensions into a standardized channel count. This mapping allows for consistent integration and fair comparison across baseline architectures with varying design specifications.

## G   BASELINE

We first examine three supervised decoding approaches employing diverse backbone architectures without pre-training (Jing et al., 2023; Li et al., 2022; Song et al., 2021). Additionally, we include two methods utilizing either contrastive learning or masked modeling for pre-training, specifically tailored to single-task decoding scenarios, enabling generic feature extraction prior to downstream decoding tasks (Woo et al., 2022; Wu et al., 2022a). Notably, these baseline approaches exclusively support single-subject and single-task decoding. Subsequently, we explore representative large-scale modeling techniques that integrate neural recordings across multiple subjects or tasks. Specifically, we discuss BIOT (Yang et al., 2023), H2DiLR (Wu et al., 2025), and LaBraM (Jiang et al., 2024), which are general-purpose neural representation learning frameworks pre-trained on extensive EEG data. While these frameworks perform pre-training using multi-subject and multi-task data, they necessitate downstream fine-tuning for specific single-task decoding scenarios. Finally, we consider

Table A2: Detailed architecture specifications for models utilized in this work.

| Architecture | Hyperparameters | Values |
|---|---|---|
| Tokenizer | Number of filters | {8, 16, 16, 32, 64} |
| | Kernel size | {15, 7, 5, 3, 3} |
| | Stride | {7, 4, 3, 2, 2} |
| Brain-regional MoE | Number of Blocks | 4 |
| | Hidden Size | 64 |
| | MLP Size | 128 |
| | Number of Head | 8 |
| Prediction Head | Number of Layer | 2 |
| | Hidden Size | 256 |
| Reconstructor | Number of Layer | 2 |
| | Hidden Size | 256 |

NEUROLM (Jiang et al., 2025), a multi-task neural decoding framework that formulates neural decoding as a question-answering task, leveraging large language models (LLMs) to facilitate simultaneous decoding across multiple tasks.

Table A3: Task instruction designed for NeuroLM implementation on our tasks.

| Task | Instruction Description |
|---|---|
| Initial Decoding | [SEP] Question: Which initial is being articulated? Options: (A): 'b', (B): 'p', (C): 'm', (D): 'f', (E): 'd', (F): 't', (G): 'n', (H): 'l', (I): 'g', (J): 'h', (K): 'j', (L): 'q', (M): 'x', (N): 'z', (O): 'c', (P): 's', (Q): 'zh', (R): 'ch', (S): 'sh', (T): 'r', (U): 'w', (V): 'y', (W): 'k'. Answer: { (A), (B), (C), (D), (E), (F), (G), (H), (I), (J), (K), (L), (M), (N), (O), (P), (Q), (R), (S), (T), (U), (V), (W)} [END] |
| Final Decoding | [SEP] Question: Which final cluster is being articulated? Options: (A): 'a', 'ua', 'ia'. (B): 'uai', 'uan'. (C): 'an', 'ai'. (D): 'en', 'eng', 'un'. (E): 'ing', 'in', 'ian', 'ie'. (F): 'iu', 'iao', 'iang'. (G): 'ui', 'ei'. (H): 've', 'ue', 'v', 'i'. (I): 'o', 'u', 'uo', 'ou'. (J): 'iong', 'ong'. (K): 'ang', 'ao', 'uang'. (L): 'e', 'er'. Answer: {(A), (B), (C), (D), (E), (F), (G), (H), (I), (J), (K), (L)} [END] |
| Tone Decoding | [SEP] Question: Which tone is being articulated? Options: (A): 'First'. (B): 'Second' (C): 'Third' (D): 'Fourth'. Answer: {(A), (B), (C), (D), (E), (F), (G), (H), (I), (J), (K), (L)} [END] |
| Seizure Detection | Question: Is this EEG segment ictal? Answer: {Yes, No} [END] |
| Seizure Prediction | Question: Is this EEG segment pre-ictal? Answer: {Yes, No} [END] |

## H IMPLEMENTATION DETAILS

All experiments were performed on GPU workstations equipped with NVIDIA H800 GPUs, utilizing PyTorch-based implementations. For baseline methods with publicly available code (Woo et al., 2022; Eldele et al., 2021; Wu et al., 2022a; Yang et al., 2023; Jiang et al., 2025), we reproduced their experimental results by strictly following the official implementations and configurations provided by the original authors. Specifically, we crafted the task instructions detailed in Tab. A3 to facilitate the implementation of NeuroLM. For baselines lacking publicly released implementations (Jing et al., 2023; Li et al., 2022; Song et al., 2021), we employed the re-implementations offered within the BOIT framework (Yang et al., 2023). Specifically, baseline approaches without pre-training were trained for a fixed duration of 200 epochs. For optimization, we used the AdamW optimizer (Loshchilov & Hutter, 2019) with a base learning rate of 5e-4, momentum parameters of $\beta_1, \beta_2 = 0.9, 0.999$, and a weight decay of 0.01. The learning rate decayed following a cosine annealing schedule. Training employed a batch size equivalent to eight times the number of subjects, and a dropout probability

Table A4: Ablation on scaling-up network parameters (embedding dimension and blocks).

| Architecture | Initial Decoding | Final Decoding | Tone Decoding | Seizure Prediction | Seizure Detection |
|---|---|---|---|---|---|
| Transformer-4-64 | **0.2901** | 0.3161 | 0.4341 | 0.7877 | 0.8957 |
| Transformer-4-128 | 0.2881 | **0.3337** | **0.4384** | 0.7923 | 0.8909 |
| Transformer-8-64 | 0.2846 | 0.3258 | 0.4348 | **0.8032** | **0.9011** |
| Transformer-16-512 | 0.1597 | 0.2208 | 0.3216 | 0.6393 | 0.7059 |

of 0.1 was consistently applied. For baseline models incorporating pre-training, we strictly adhered to the pre-training setups and experimental protocols specified by their original authors. During the fine-tuning phase for these pre-trained methods, we utilized the same exact training configuration as detailed above. We now describe the pre-training procedures of the proposed Neuro-MoBRE model. During our regional masked autoencoding pre-training phase, optimization was conducted using the AdamW (Loshchilov & Hutter, 2019) optimizer with a base learning rate of 5e-5, momentum parameters $\beta_1, \beta_2 = 0.9, 0.95$, and weight decay set to 0.05. The learning rate was scheduled via cosine annealing over a fixed training duration of 800 epochs, with a batch size of 32 for each task. Additionally, "co-upcycling" was implemented by pruning 50% of parameters having the lowest magnitudes. Following pre-training, our multi-task neural decoding stage consisted of training for a fixed total of 200 epochs, employing a batch size equal to four per subject per task. In this phase, we used the AdamW optimizer, setting the base learning rate to 5e-4, momentum parameters $\beta_1, \beta_2 = 0.9, 0.95$, and a higher weight decay of 0.1.

# I    ADDITIONAL ABLATION STUDY

## I.1    ABLATION ON NETWORK PARAMETERS.

We further investigate the scaling behavior of our model by systematically varying the embedding dimension and number of layers (blocks). Neuro-MoBRE-N-D denotes a model configuration consisting of $N$ layers and embedding dimension $D$, with Transformer-4-64 serving as the baseline configuration. The average top-1 accuracy is reported, and we follow the same experimental setup as described in Appendix H. As illustrated in Tab. A4, modest increases in network depth and embedding dimension consistently yield improvements in decoding performance. However, excessively scaling the model size, such as utilizing 16 blocks combined with a 512-dimensional embedding, results in a marked degradation in performance, indicative of model collapse. We hypothesize this phenomenon arises due to insufficient training data relative to the increased number of model parameters. Such an observation aligns with our assertion that, given an effective resolution of data heterogeneity, excessively large models are not required to achieve high-quality representations in multi-subject, multi-task neurophysiological decoding scenarios.

# J    BROADER IMPACT

The Neuro-MoBRE model proposed in this study holds significant potential implications across scientific, clinical, and societal domains. By explicitly addressing data heterogeneity challenges inherent to neurophysiological decoding, Neuro-MoBRE substantially enhances the robustness and precision of brain-computer interfaces (BCIs). This advancement contributes directly to fundamental neuroscience research, offering deeper insights into neural network organization and functional dynamics across different brain regions and tasks.

Clinically, Neuro-MoBRE's ability to accurately and reliably decode intracranial neurophysiological signals across diverse subjects and tasks has crucial implications. Improved decoding accuracy can substantially enhance diagnostic precision, particularly in detecting epileptic seizures, thereby facilitating timely and effective medical interventions. Moreover, decoding complex neural activities such as language production and motor functions can significantly advance rehabilitation strategies, improve prosthetic device control for patients with motor impairments, and enhance communication assistance technologies for individuals experiencing speech difficulties.

Table A5: The first set of 407 Mandarin Chinese characters used as reading material, with corresponding syllables organized by initial phonetic order. For infrequently used characters, participants were presented directly with the syllable to pronounce.

| Characters | Syllables | Characters | Syllables | Characters | Syllables |
|---|---|---|---|---|---|
| 阿 | ā | 扯 | chě | 吊 | diào |
| 爱 | ài | 趁 | chèn | 叠 | dié |
| 安 | ān | 城 | chéng | 顶 | dǐng |
| 昂 | áng | 痴 | chī | 丢 | diū |
| 袄 | ǎo | 虫 | chóng | 东 | dōng |
| 拔 | bá | 愁 | chóu | 豆 | dòu |
| 白 | bái | 初 | chū | 读 | dú |
| 板 | bǎn | chuā | chuā | 短 | duǎn |
| 帮 | bāng | 揣 | chuāi | 对 | duì |
| 保 | bǎo | 穿 | chuān | 蹲 | dūn |
| 杯 | bēi | 床 | chuáng | 夺 | duó |
| 本 | běn | 吹 | chuī | 鹅 | é |
| 崩 | bēng | 春 | chūn | 恩 | ēn |
| 鼻 | bí | 戳 | chuō | ēng | ēng |
| 编 | biān | 次 | cì | 耳 | ěr |
| 标 | biāo | 葱 | cōng | 罚 | fá |
| 瘪 | biě | 凑 | còu | 反 | fǎn |
| 宾 | bīn | 粗 | cū | 方 | fāng |
| 病 | bìng | 窜 | cuàn | 肥 | féi |
| 伯 | bó | 脆 | cuì | 粉 | fěn |
| 补 | bǔ | 存 | cún | 风 | fēng |
| 擦 | cā | 搓 | cuō | 福 | fú |
| 财 | cái | 打 | dǎ | 否 | fǒu |
| 残 | cán | 带 | dài | 付 | fù |
| 仓 | cāng | 胆 | dǎn | 尬 | gà |
| 槽 | cáo | 党 | dǎng | 盖 | gài |
| 测 | cè | 到 | dào | 敢 | gǎn |
| 参 | cān | 德 | dé | 缸 | gāng |
| 层 | céng | 得 | děi | 告 | gào |
| 茶 | chá | 扽 | dèn | 割 | gē |
| 柴 | chái | 等 | děng | 给 | gěi |
| 产 | chǎn | 底 | dǐ | 根 | gēn |
| 唱 | chàng | 爹 | diē | 耕 | gēng |
| 超 | chāo | 电 | diàn | 共 | gòng |

Table A6: The second set of 407 Mandarin Chinese characters used as reading material, with corresponding syllables organized by initial phonetic order.

| Characters | Syllables | Characters | Syllables | Characters | Syllables |
|---|---|---|---|---|---|
| 狗 | gǒu | 金 | jīn | 冷 | lěng |
| 估 | gū | 镜 | jìng | 力 | lì |
| 瓜 | guā | 窘 | jiǒng | 俩 | liǎ |
| 怪 | guài | 酒 | jiǔ | 连 | lián |
| 关 | guān | 举 | jǔ | 良 | liáng |
| 广 | guǎng | 捐 | juān | 料 | liào |
| 贵 | guì | 决 | jué | 列 | liè |
| 滚 | gǔn | 俊 | jùn | 林 | lín |
| 裹 | guǒ | 卡 | kǎ | 领 | lǐng |
| 哈 | hā | 开 | kāi | 柳 | liǔ |
| 孩 | hái | 砍 | kǎn | 龙 | lóng |
| 寒 | hán | 糠 | kāng | 楼 | lóu |
| 航 | háng | 靠 | kào | 路 | lù |
| 耗 | hào | 科 | kē | 卵 | luǎn |
| 河 | hé | 克 | kè | 轮 | lún |
| 黑 | hēi | 肯 | kěn | 罗 | luó |
| 恨 | hèn | 坑 | kēng | 滤 | lü |
| 恒 | héng | 孔 | kǒng | 略 | lüè |
| 烘 | hōng | 口 | kǒu | 马 | mǎ |
| 吼 | hǒu | 库 | kù | 买 | mǎi |
| 虎 | hǔ | 夸 | kuā | 慢 | màn |
| 画 | huà | 快 | kuài | 忙 | máng |
| 坏 | huài | 款 | kuǎn | 毛 | máo |
| 换 | huàn | 狂 | kuáng | 美 | měi |
| 慌 | huāng | 亏 | kuī | 门 | mén |
| 悔 | huǐ | 捆 | kǔn | 猛 | měng |
| 昏 | hūn | 阔 | kuò | 米 | mǐ |
| 货 | huò | 拉 | lā | 面 | miàn |
| 机 | jī | 来 | lái | 苗 | miáo |
| 价 | jià | 懒 | lǎn | 灭 | miè |
| 尖 | jiān | 浪 | làng | 民 | mín |
| 桨 | jiǎng | 老 | lǎo | 命 | mìng |
| 交 | jiāo | 勒 | lēi | 谬 | miù |
| 姐 | jiě | 雷 | léi | 魔 | mó |

Table A7: The third set of 407 Mandarin Chinese characters used as reading material, with corresponding syllables organized by initial phonetic order. For infrequently used characters, participants were presented directly with the syllable to pronounce.

| Characters | Syllables | Characters | Syllables | Characters | Syllables |
|---|---|---|---|---|---|
| 谋 | móu | 盆 | pén | 乳 | rǔ |
| 木 | mù | 碰 | pèng | ruá | ruá |
| 拿 | ná | 皮 | pí | 软 | ruǎn |
| 奶 | nǎi | 偏 | piān | 锐 | ruì |
| 男 | nán | 票 | piào | 润 | rùn |
| 囊 | náng | 瞥 | piē | 弱 | ruò |
| 闹 | nào | 品 | pǐn | 洒 | sǎ |
| 讷 | nè | 瓶 | píng | 赛 | sài |
| 内 | nèi | 破 | pò | 伞 | sǎn |
| 嫩 | nèn | 剖 | pōu | 嗓 | sǎng |
| 能 | néng | 普 | pǔ | 骚 | sāo |
| 泥 | ní | 骑 | qí | 涩 | sè |
| 年 | nián | 掐 | qiā | 森 | sēn |
| 娘 | niáng | 浅 | qiǎn | 僧 | sēng |
| 尿 | niào | 枪 | qiāng | 沙 | shā |
| 捏 | niē | 桥 | qiáo | 筛 | shāi |
| 您 | nín | 窃 | qiè | 闪 | shǎn |
| 凝 | níng | 琴 | qín | 伤 | shāng |
| 牛 | niú | 情 | qíng | 烧 | shāo |
| 农 | nóng | 穷 | qióng | 赊 | shē |
| 耨 | nòu | 球 | qiú | 谁 | shuí |
| 奴 | nú | 取 | qǔ | 神 | shén |
| 暖 | nuǎn | 劝 | quàn | 绳 | shéng |
| 挪 | nuó | 缺 | quē | 石 | shí |
| 女 | nü | 群 | qún | 手 | shǒu |
| 虐 | nüè | 然 | rán | 书 | shū |
| 哦 | ó | 让 | ràng | 刷 | shuā |
| 藕 | ǒu | 饶 | ráo | 帅 | shuài |
| 爬 | pá | 热 | rè | 栓 | shuān |
| 牌 | pái | 忍 | rěn | 爽 | shuǎng |
| 盘 | pán | 扔 | rēng | 水 | shuǐ |
| 旁 | páng | 日 | rì | 顺 | shùn |
| 抛 | pāo | 容 | róng | 硕 | shuò |
| 赔 | péi | 肉 | ròu | 死 | sǐ |

Table A8: The fourth set of 407 Mandarin Chinese characters used as reading material, with corresponding syllables organized by initial phonetic order.

| Characters | Syllables | Characters | Syllables | Characters | Syllables |
|---|---|---|---|---|---|
| 松 | sōng | 午 | wǔ | 早 | zǎo |
| 搜 | sōu | 洗 | xǐ | 则 | zé |
| 酥 | sū | 霞 | xiá | 贼 | zéi |
| 算 | suàn | 险 | xiǎn | 怎 | zěn |
| 岁 | suì | 向 | xiàng | 增 | zēng |
| 损 | sǔn | 笑 | xiào | 渣 | zhā |
| 锁 | suǒ | 写 | xiě | 债 | zhài |
| 塔 | tǎ | 心 | xīn | 展 | zhǎn |
| 抬 | tái | 形 | xíng | 涨 | zhǎng |
| 谈 | tán | 熊 | xióng | 找 | zhǎo |
| 躺 | tǎng | 修 | xiū | 遮 | zhē |
| 讨 | tǎo | 许 | xǔ | 这 | zhè |
| 特 | tè | 选 | xuǎn | 针 | zhēn |
| 藤 | téng | 学 | xué | 蒸 | zhēng |
| 替 | tì | 寻 | xún | 直 | zhí |
| 田 | tián | 芽 | yá | 肿 | zhǒng |
| 条 | tiáo | 烟 | yān | 州 | zhōu |
| 铁 | tiě | 养 | yǎng | 煮 | zhǔ |
| 停 | tíng | 药 | yào | 抓 | zhuā |
| 桶 | tǒng | 野 | yě | 拽 | zhuāi |
| 偷 | tōu | 衣 | yī | 砖 | zhuān |
| 图 | tú | 银 | yín | 撞 | zhuàng |
| 团 | tuán | 鹰 | yīng | 追 | zhuī |
| 腿 | tuǐ | 哟 | yō | 准 | zhǔn |
| 吞 | tūn | 永 | yǒng | 捉 | zhuō |
| 拖 | tuō | 油 | yóu | 字 | zì |
| 挖 | wā | 雨 | yǔ | 总 | zǒng |
| 外 | wài | 元 | yuán | 走 | zǒu |
| 万 | wàn | 月 | yuè | 组 | zǔ |
| 忘 | wàng | 云 | yún | 钻 | zuān |
| 围 | wéi | 杂 | zá | 醉 | zuì |
| 文 | wén | 栽 | zāi | 尊 | zūn |
| 翁 | wēng | 暂 | zàn | 左 | zuǒ |
| 窝 | wō | 葬 | zàng | | |

From an applicability and accessibility standpoint, the generalized decoding capabilities of Neuro-MoBRE across multiple tasks and subjects have the potential to significantly expand the accessibility of neurotechnological solutions. Improved cross-subject decoding performance on previously unseen individuals greatly reduces system calibration requirements, allowing more effective personalization and broader practical deployment.

Nevertheless, despite the promising advancements presented, several ethical and societal considerations must be addressed. Enhanced decoding capabilities pose risks concerning patient privacy and data security, necessitating rigorous ethical guidelines, robust data protection measures, and transparent informed consent protocols. Additionally, the increased accuracy and potential widespread adoption of BCIs could lead to misuse or unauthorized surveillance, highlighting the need for comprehensive policy frameworks, advanced encryption methods, and clear guidelines for responsible usage. Finally, the current reliance on invasive intracranial data collection methods introduces inherent procedural risks. Future research should prioritize validating similar models with high-quality non-invasive data, reducing invasiveness, and improving safety and accessibility for broader populations.

## K   DATA AVAILABILITY

Data access will be provided to qualified researchers upon reasonable request to the corresponding author, subject to standard data use agreements to ensure ethical and legal compliance.

