# OpenReview forum: "Neuro-MoBRE: Exploring Multi-subject Multi-task Intracranial Decoding via Explicit Heterogeneity Resolving"
_ICLR.cc/2026/Conference — Submitted to ICLR 2026_

### Official Review · Reviewer_vjm2 · 2025-10-15

**Soundness:** 2
**Presentation:** 3
**Contribution:** 2
**Rating:** 4
**Confidence:** 3

**Summary:**

The authors propose Neural Mixture of Brain Regional Experts (Neuro-MoBRE) as a general-purpose neurophysiological decoding framework, which uses intracranial data collected from 11 subjects across five distinct decoding tasks. Neuro-MoBRE incorporates a brain-regional-temporal embedding mechanism within a decoder-only transformer architecture to effectively handle structural heterogeneity. It addresses the critical concern of data heterogeneity and low signal-to-noise ratio (SNR) of neurophysiological signals.

**Strengths:**

1. The paper concludes two valuable and unresolved challenges: (1) Ubiquitous data heterogeneity in neurophysiological modeling and (2) Semantic vagueness and low signal-to-noise ratio of neurophysiological signals.

2. Neuro-MoBRE outperforms compared methods in two tasks. The authors also conduct ablation study to show the effectiveness of each part of the model.

**Weaknesses:**

1. Typos: Figure A1 and Table A1 in page 6 link to wrong figure and table.

2. The paper lacks experiments on publicly available datasets. For example, Labram conducts experiments on TUAB and TUEV.

3. While many existing works have achieved sentence-level language decoding (eeg-to-text, fmri-to-text, etc.), the proposed Neuro-MoBRE still only focuses on character-level decoding.

4. The paper lacks a case analysis part to show the detailed content of decoding results.

5. While the authors claim to perform multi-task decoding, the experiment only cover Language decoding task and Epileptic seizure diagnosis tasks, which weakens the effectiveness of the framework.

**Questions:**

1. I notice that there's no submitted code, which largely affects the potential influence of this paper. Will the authors open-source this framework?

2. In section 3.1, while the order of the framework is Brain-regional-temporal Tokenization, Task-disentangled Information Aggregation, and Brain-regional MoE according to figure 1, why the authors introduce Brain-regional MoE part before Task-disentangled Information Aggregation?

3. Can Neuro-MoBRE outperforms other compared models when the setting is not multi-subject multi-task?

**Details Of Ethics Concerns:**

human subjects involved

---

> ### Author Response · Authors · 2025-11-14
> **Response to Reviewer vjm2 (1/2)**
>
> ## Response to Reviewer vjm2
> ---
>
> # Reply to Weakness
>
> > **W1: Typos: Figure A1 and Table A1 in page 6 link to wrong figure and table.**
>
> **R**: We thank the reviewer for pointing out the incorrect links to Figure A1 and Table A1 on page 6. We have corrected the links in the revised manuscript to point to the intended figure and table.
>
> > **W2: The paper lacks experiments on publicly available datasets. For example, Labram conducts experiments on TUAB and TUEV.**
>
> **R**: We thank the reviewer for highlighting this point. Our study specifically targets multi-task, multi-subject **intracranial decoding**, for which **no available public datasets** currently exist. Merging multiple publicly available non-invasive datasets (e.g., EEG) often introduces substantial variability due to differences in equipment, acquisition protocols, and subject cohorts, which can confound the evaluation of methods for handling task-related and subject-specific heterogeneity.
>
> To mitigate unnecessary heterogeneity, we collected intracranial recordings from the same participant cohort using identical equipment and protocols, and carried out five distinct decoding tasks for each of 11 participants, including language decoding and epileptic seizure diagnosis. In contrast, TUAB and TUEV primarily involve detecting abnormal EEG states, tasks that are relatively with minimal cognitive demands and often separable by amplitude or frequency features. Our tasks involve cognitive and semantic decoding (e.g., phoneme classification with 23 classes), which exhibits greater inter-subject heterogeneity and better demonstrates the importance of our heterogeneity-handling framework in a multi-task, multi-subject invasive recording setting.
>
> > **W3: While many existing works have achieved sentence-level language decoding (eeg-to-text, fmri-to-text, etc.), the proposed Neuro-MoBRE still only focuses on character-level decoding.**
>
> **R**: We appreciate the reviewer's comment and would like to clarify our contributoin. The primary objective of our work is to propose a **generic multi-task, multi-subject intracranial decoding framework** that explicitly addresses the heterogeneity arising from biological differences across subjects, as well as variability in experimental setups and conditions. Our contribution centers on developing a general decoding framework rather than focusing solely on sentence-level language decoding.
>
> To demonstrate the flexibility of our framework, we use phoneme-level decoding tasks and seizure diagnostic tasks as proof-of-concept applications. While many existing non-invasive language-decoding approaches, such as EEG [1] and fMRI [2], achieve sentence-level decoding, they often rely on the **expressive capacity of pre-trained large language models in a retrieval-based fashion**, where neural representations are mapped into the embedding space of the LLM and the decoded sentence is retrieved. In contrast, our work performs direct phoneme-level decoding, following approaches such as Metzger et al. [3] and Willett et al. [4].
>
> From the decoded phoneme sequences, sentence-level decoding can readily be achieved using post-processing with language models [3,4] (e.g., n-gram models). Our pipeline can therefore be extended to sentence-level decoding, but implementing and evaluating these additional components would require substantial engineering effort, which is beyond the scope of this paper. We believe our direct phoneme decoding results nonetheless demonstrate the effectiveness and extensibility of the proposed framework.
>
> # References
>
> [1] Liu, Hanwen, et al. "Eeg2text: Open vocabulary eeg-to-text decoding with eeg pre-training and multi-view transformer." arXiv preprint arXiv:2405.02165 (2024).
>
> [2] Qiu, Weikang, et al. "MindLLM: A Subject-Agnostic and Versatile Model for fMRI-to-Text Decoding." arXiv preprint arXiv:2502.15786 (2025).
>
> [3] Metzger S L, Littlejohn K T, Silva A B, et al. A high-performance neuroprosthesis for speech decoding and avatar control[J]. Nature, 2023, 620(7976): 1037-1046.
>
> [4] Willett F R, Kunz E M, Fan C, et al. A high-performance speech neuroprosthesis[J]. Nature, 2023, 620(7976): 1031-1036.
>
> > **W4: The paper lacks a case analysis part to show the detailed content of decoding results.**
>
> **R**: We thank the reviewer for raising this point. We understand case analysis to refer to qualitative examples similar to sentence-level decoding demonstrations in EEG-to-text or fMRI-to-text studies. As stated in our previous response, our framework produces phoneme-level decoding outputs, which are essentially classification results and do not directly yield fluent sentences.

---

> ### Author Response · Authors · 2025-11-14
> **Response to Reviewer vjm2 (2/2)**
>
> > **W5: While the authors claim to perform multi-task decoding, the experiment only cover Language decoding task and Epileptic seizure diagnosis tasks, which weakens the effectiveness of the framework.**
>
> **R**: We agree that incorporating more decoding tasks could further demonstrate the versatility of our framework. However, due to the constraints of collecting intracranial brain recordings in clinical settings, it is challenging to design and record multiple cognitive tasks from a large corhort of subjects. Existing intracranial decoding endeavors [1,2,3] are mainly focused on **single task decoding within a single test subject**. In our work, we managed to collect three phoneme level decoding tasks as well as epileptic seizure diagnosis tasks within the **limited two-week monitoring window**. This required all tasks to be performed during inpatient monitoring for epilepsy surgery evaluation.
>
> Systematic performance comparisons to prior work demonstrate the effectiveness of our framework under realistic clinical constraints. To the best of our knowledge, this is **the first multi-task, multi-subject intracranial decoding study**, and we will make this dataset publicly available upon acceptance. While additional tasks would further illustrate generality, we believe our results already validate the effectiveness of the proposed framework and provide a significant data and methodological contribution to the community.
>
> # References
>
> [1] Metzger S L, Littlejohn K T, Silva A B, et al. A high-performance neuroprosthesis for speech decoding and avatar control[J]. Nature, 2023, 620(7976): 1037-1046.
>
> [2] Willett F R, Kunz E M, Fan C, et al. A high-performance speech neuroprosthesis[J]. Nature, 2023, 620(7976): 1031-1036.
>
> [3] Bouton, Chad E., et al. "Restoring cortical control of functional movement in a human with quadriplegia." Nature 533.7602 (2016): 247-250.
>
> # Reply to Question
>
> > **Q1:I notice that there's no submitted code, which largely affects the potential influence of this paper. Will the authors open-source this framework?**
>
> **R**: We plan to release the code upon acceptance of this paper. In addition, we will make the dataset available upon reasonable request to the corresponding authors, in accordance with all applicable ethical and privacy regulations. We believe that releasing both the code and data will substantially benefit the study and development of large-scale intracranial neurophysiological models and foster further research in this domain.
>
> > **Q2: In section 3.1, while the order of the framework is Brain-regional-temporal Tokenization, Task-disentangled Information Aggregation, and Brain-regional MoE according to figure 1, why the authors introduce Brain-regional MoE part before Task-disentangled Information Aggregation?**
>
> **R**: We thank the reviewer for highlighting this ordering question. The three components of our framework address different aspects of heterogeneity. We presented Brain-regional MoE before Task-disentangled Information Aggregation because the Brain-regional MoE directly builds upon the tokenization stage and is essential for constructing the overall framework. Understanding this token routing is a prerequisite for explaining how aggregated representations from these experts are subsequently disentangled across tasks. We believe this ordering offers readers a clearer conceptual flow of the framework.
>
> > **Q3: Can Neuro-MoBRE outperforms other compared models when the setting is not multi-subject multi-task?**
>
> **R**: We thank the reviewer for raising this interesting question. The three core components of our framework are each designed to mitigate specific forms of heterogeneity encountered in the multi-subject, multi-task setting:
> * Brain-regional-temporal tokenizer and regional masked autoencoding primarily address multi-subject heterogeneity.
> * Brain-regional MoE and Task-disentangled Information Aggregation primarily address multi-task heterogeneity.
>
> In a single-subject, single-task scenario, these targeted components would not confer their usual advantages, and the framework would effectively reduce to a transformer backbone with masked modeling pre-training, similar to Labram [1] and Neuro-BERT [2]. We therefore expect performance in such settings to be at least comparable to existing state-of-the-art approaches, and potentially better, due to the robust tokenization and pre-training schemes that remain part of the architecture.
>
>
> # References
>
> [1] Jiang, Wei-Bang, Li-Ming Zhao, and Bao-Liang Lu. "Large brain model for learning generic representations with tremendous EEG data in BCI." arXiv preprint arXiv:2405.18765 (2024).
>
> [2] Wu, Di, et al. "Neuro-bert: Rethinking masked autoencoding for self-supervised neurological pretraining." arXiv preprint arXiv:2204.12440 (2022).

---

> ### Author Response · Authors · 2025-11-24
>
> Dear Reviewer vjm2,
>
> Thank you for your thoughtful review and for raising important points regarding our work. We have provided detailed responses to each of your comments in the rebuttal.
>
> We would greatly appreciate it if you could revisit our responses when you have time. Please let us know if any concerns remain or if further clarification would help. If you feel that our rebuttal satisfactorily addresses your questions, we kindly ask you to consider reflecting this in your overall score.
>
> We value your feedback and willingness to engage in the discussion, and we thank you again for helping us improve the manuscript.
>
> Authors

---

> ### Comment · Reviewer_vjm2 · 2025-11-25
>
> Thanks for the authors' reply. As no additional experiments were conducted, I will retain my initial rating.

---

> > ### Author Response · Authors · 2025-11-25
> >
> > Our proposed Neuro‑MoBRE is a **multi-task, multi-subject decoding framework**, whereas most existing works are fundamentally a **single-task model** such as LaBraM [1]. Although LaBraM is pre‑trained on multiple datasets, it requires fine‑tuning on a single dataset for a specific task. In contrast, Neuro‑MoBRE is designed to handle multiple decoding tasks within one unified framework.
> >
> > NeuroLM [2], another multi-task model, is conceptually more similar to Neuro‑MoBRE. However, the inferior performance of NeuroLM compared to LaBraM on TUAB and TUEV, a drop of roughly **20% in balanced accuracy**, suggests that **heterogeneity across datasets** (differences in equipment, acquisition protocols, and subject cohorts) makes true multi-task decoding difficult when merging existing public EEG datasets. **This is a key reason we chose to curate a controlled, homogeneous intracranial dataset for our study**.
> >
> > Following the reviewer’s suggestion, we nevertheless conducted experiments on TUAB and TUEV, comparing Neuro‑MoBRE, NeuroLM, and LaBraM. Because our RMAE module assumes that the **same cohort of subjects performs the same set of neural tasks**, which does not hold for TUAB/TUEV, we modified RMAE to a standard masked autoencoding during pre‑training.
> >
> > **Performance comparison on TUAB:**
> >
> > | Method        | Multi‑task | Balanced Acc      | AUROC             |
> > |:-------------:|:----------:|:-----------------:|:-----------------:|
> > | LaBraM [1]    | No         | 0.8140 ± 0.0019   | 0.9022 ± 0.0009   |
> > | NeuroLM [2]   | Yes        | 0.7826 ± 0.0065   | 0.6975 ± 0.0081   |
> > | **Neuro‑MoBRE** | Yes        | **0.8199 ± 0.0056** | **0.8873 ± 0.0062** |
> >
> > **Performance comparison on TUEV:**
> >
> > | Method        | Multi‑task | Balanced Acc      | Cohen’s Kappa     |
> > |:-------------:|:----------:|:-----------------:|:-----------------:|
> > | LaBraM [1]    | No         | 0.6409 ± 0.0065   | 0.6637 ± 0.0093   |
> > | NeuroLM [2]   | Yes        | 0.4560 ± 0.0048   | 0.4285 ± 0.0048   |
> > | **Neuro‑MoBRE** | Yes        | **0.5580 ± 0.0076** | **0.5428 ± 0.0089** |
> >
> > From these results:
> > - NeuroLM (multi-task) performs significantly worse than LaBraM (single-task), particularly on TUEV (~18% drop in balanced accuracy).
> > - Neuro‑MoBRE outperforms NeuroLM by a large margin on both datasets, indicating better heterogeneity handling despite still trailing LaBraM.
> >
> > These findings support our position: merging heterogeneous EEG datasets limits the potential of multi-task frameworks, and a curated, controlled dataset, as used in our main experiments, is essential for fair and effective multi-task evaluation.
> >
> > ---
> >
> > **References**
> > [1] Jiang, W.-B., Zhao, L.-M., & Lu, B.-L. *Large brain model for learning generic representations with tremendous EEG data in BCI*. arXiv:2405.18765 (2024).
> >
> > [2] Jiang, W., Wang, Y., Lu, B.-L., & Li, D. *NeuroLM: A universal multi-task foundation model for bridging the gap between language and EEG signals*. ICLR 2025.
> >
> > ---

---

> > ### Author Response · Authors · 2025-11-27
> > **Follow-Up on Rebuttal Response**
> >
> > Dear Reviewer vjm2,
> >
> > We sincerely appreciate your detailed comments about evaluation consistency and dataset selection. In our rebuttal, we conducted additional experiments on the TUAB and TUEV datasets, compared Neuro‑MoBRE with LaBraM and NeuroLM, and clearly explained why our curated intracranial dataset provides a controlled, homogeneous benchmark for multi‑task, multi‑subject decoding.
> >
> > We hope these new results and explanations address your concerns. Please let us know if any questions remain or if further clarification is needed. If you find the added analysis satisfactory, we would be grateful if you could update your assessment accordingly.
> >
> > Thank you again for your helpful and thorough review.
> >
> > Authors

---

### Official Review · Reviewer_ebUX · 2025-10-30

**Soundness:** 3
**Presentation:** 2
**Contribution:** 2
**Rating:** 2
**Confidence:** 4

**Summary:**

The paper proposes Neuro-MoBRE, a novel modular mixture-of-experts (MoE) framework designed for neural decoding across multiple brain regions and tasks. The method assigns region-specific experts to distinct brain areas and employs a gating mechanism to adaptively integrate their outputs for robust behavioral prediction. The authors evaluate the model on intracranial recordings from 11 subjects across five decoding tasks under both within-subject and cross-subject settings, demonstrating improved within-subject performance over baseline models and the potential for cross-subject generalization.

**Strengths:**

1. The proposed framework achieved the best within-subject performance across all decoding tasks compared with the baselines.
2. The proposed Neuro-MoBRE framework is conceptually interesting, especially its modular mixture-of-experts (MoE) design that allocates region-specific experts for decoding across multiple brain areas.

**Weaknesses:**

Major concerns:
1. In the introduction, the authors identify low SNR in non-invasive neurophysiological recordings as one of the main challenges motivating this work (line 70). However, the paper states:
***“To circumvent the limitations posed by low SNRs in non-invasive neurophysiological recordings, we rigorously evaluate Neuro-MoBRE using intracranial data collected from 11 subjects across five distinct decoding tasks.”*** If the primary goal is to enhance robustness to low-SNR conditions, evaluating exclusively on high-SNR intracranial data does not convincingly demonstrate such robustness or the model’s applicability to non-invasive modalities. While using intracranial data as a clean benchmark to establish an upper bound of performance is reasonable, the paper should make this rationale explicit and avoid implying that this setup “circumvents” the low-SNR limitation, as the current experiments do not address or validate performance under low-SNR conditions. The authors could either include experiments using low-SNR modalities (e.g., EEG or MEG) to empirically test robustness, or clearly reframe the motivation to indicate that the intracranial evaluation serves only as a controlled, high-SNR benchmark. Based on the current paper, I don't see the proposed architecture/framework addressing the low SNR issue.

2. The paper claimed they achieve “zero-shot generalization". Typically, zero-shot refers to scenarios involving unseen datasets probably with different demographics, acquisition setups, or task domains. For instance, directly testing the model on unseen data collected from a separate study. The scenario described in the paper appears to involve unseen subjects within the same dataset, which would be more accurately characterized as cross-subject generalization rather than zero-shot generalization?

3. The model seems heavily tailored to the specific dataset (with specific tasks) used in the study, which raises questions about scalability. The use of subject-wise models for RMAE sessions may become computationally prohibitive as the dataset or population size increases. The term “generalization” should therefore be used with caution, especially since no external or cross-dataset experiments are provided.

4. For the unseen subject decoding performance, the paper didn't compare with other baselines.


Minor suggestions:
- Some tables (e.g., Table 3 and 4) are missing standard deviation (std) values, while Tables 1 and 2 include them. Consistent reporting of mean ± std would improve clarity and comparability.
- The description of the proposed method is sometimes unclear, and several components would benefit from improved clarity and organization.

**Questions:**

1. If I understand correctly, each decoding task is said to have its own set of classification tokens within this multi-task model. How does the model handle new, unseen tasks during inference? Is there a mechanism for extending the task token space without retraining the entire model?
2. When comparing against baselines such as PopT that are also trained on intracranial recordings, are these baselines fine-tuned from their pre-trained weights, or trained from scratch?
3. Given that prior works such as PopT have already tackled the challenge of inter-subject electrode placement variability, what is the key novelty or improvement introduced by the proposed method in this aspect?
4. What distinguishes the proposed brain-regional MoE from a standard MoE architecture (from the architecture aspect)?
5. Will the dataset be released publicly if the paper is accepted?

---

> ### Author Response · Authors · 2025-11-14
> **Response to Reviewer ebUX  (1/3)**
>
> ## Response to Reviewer ebUX
> ---
>
> # Reply to Weakness
>
> > **W1: In the introduction, the authors identify low SNR in non-invasive neurophysiological recordings as one of the main challenges motivating this work (line 70). However, the paper states: 'To circumvent the limitations posed by low SNRs in non-invasive neurophysiological recordings, we rigorously evaluate Neuro-MoBRE using intracranial data collected from 11 subjects across five distinct decoding tasks'. If the primary goal is to enhance robustness to low-SNR conditions, evaluating exclusively on high-SNR intracranial data does not convincingly demonstrate such robustness or the model's applicability to non-invasive modalities. While using intracranial data as a clean benchmark to establish an upper bound of performance is reasonable, the paper should make this rationale explicit and avoid implying that this setup 'circumvents' the low-SNR limitation, as the current experiments do not address or validate performance under low-SNR conditions. The authors could either include experiments using low-SNR modalities (e.g., EEG or MEG) to empirically test robustness, or clearly reframe the motivation to indicate that the intracranial evaluation serves only as a controlled, high-SNR benchmark. Based on the current paper, I don't see the proposed architecture/framework addressing the low SNR issue.**
>
> **R**: We thank the reviewer for this comment and would like to clarify a potential misunderstanding of our claim regarding low-SNR recordings. In this work, we do not aim to directly solve the low‑SNR problem in non‑invasive neurophysiological recordings. Instead, our choice to use intracranial recordings was motivated by the need to circumvent these limitations in order to establish a controlled, high‑SNR benchmark for evaluating multi‑task, multi‑subject decoding architectures.
>
> The low SNR of EEG, caused by signal attenuation through the skull and dura mater, is known to exacerbate inter‑subject variability in higher‑level cognitive and semantic decoding tasks such as language processing. By using intracranial signals from the same participant cohort, under identical equipment and protocols, we are able to minimize this measurement‑related heterogeneity and focus our evaluation on the heterogeneity arising from biological and task differences across subjects and tasks. Our dataset includes five distinct decoding tasks (including phoneme‑level language decoding and epileptic seizure diagnosis) performed by 11 participants.
>
> As suggested, we will revise the manuscript to explicitly frame the intracranial evaluation as a controlled high‑SNR benchmark rather than as a direct test of robustness to low‑SNR conditions, thereby aligning the motivation and experimental design more clearly.
>
> > **W2: The paper claimed they achieve 'zero-shot generalization'. Typically, zero-shot refers to scenarios involving unseen datasets probably with different demographics, acquisition setups, or task domains. For instance, directly testing the model on unseen data collected from a separate study. The scenario described in the paper appears to involve unseen subjects within the same dataset, which would be more accurately characterized as cross-subject generalization rather than zero-shot generalization?**
>
> **R**: We thank the reviewer for this helpful clarification. We agree that zero-shot typically refers to evaluating a model on entirely unseen datasets or domains. In our study, the evaluation involves training on a subset of subjects (source domain) and directly testing on completely held‑out subjects (target domain), without any fine‑tuning or adaptation.
>
> Our intended meaning was 'zero‑shot transfer to unseen subjects' i.e., applying a trained model to new subjects with no subject-specific training samples. However, we agree with the reviewer that the term cross-subject generalization is more precise for this case and avoids ambiguity. We will adopt this terminology throughout the revised manuscript and clearly define our evaluation setting early in the paper.

---

> ### Author Response · Authors · 2025-11-14
> **Response to Reviewer ebUX (2/3)**
>
> > **W3: The model seems heavily tailored to the specific dataset (with specific tasks) used in the study, which raises questions about scalability. The use of subject-wise models for RMAE sessions may become computationally prohibitive as the dataset or population size increases. The term 'generalization' should therefore be used with caution, especially since no external or cross-dataset experiments are provided.**
>
> **R**: We respectfully disagree. All of our proposed modules are designed for multi-task, multi-subject decoding in a modality- and dataset-agnostic manner. Specifically:, Brain‑regional‑temporal tokenizer processes neural recordings of any channel count and signal length, ensuring adaptability to different acquisition systems. Brain‑regional MoE and Task‑disentangled Information Aggregation enable the framework to handle arbitrary neural decoding tasks, without dependence on specific dataset characteristics. Thus, the architecture is not tailored only to our current dataset but is broadly applicable to other multi-subject, multi-task intracranial or non-invasive datasets.
>
> As analyzed in Section E of the appendix, assume data are collected from m subjects, each with N samples, and the model has P parameters. Both previous EEG foudation models and our RMAE have the same complexity of O(mN⋅P). In spatial complexity, RMAE requires O(P) memory, independent of the number of subjects m. This ensures scalability even for large populations.
>
> > **W4: For the unseen subject decoding performance, the paper didn't compare with other baselines.**
>
> **R**: Our work primarily targets multi-task, multi-subject intracranial decoding, and the main performance comparisons to baselines are provided in Tables 1 and 2. The results we report for unseen subject decoding are intended as a demonstration of Neuro‑MoBRE’s ability to handle inter‑subject heterogeneity, rather than as a core benchmark. Most existing baselines do not natively support leave-one-subject-out or unseen-subject decoding without substantial adaptation, which is why we did not include their results in this setting. For fairness and reproducibility, we chose not to introduce unverified adaptations of those baselines. We will clarify this point in the revised manuscript to ensure our evaluation scope and choices are explicit.
>
> # Reply to Minor Suggestions
>
> > **S1: Some tables (e.g., Table 3 and 4) are missing standard deviation (std) values, while Tables 1 and 2 include them. Consistent reporting of mean ± std would improve clarity and comparability.**
>
> **R**: We thank the reviewer for pointing this out. We will update Tables 3 and 4 in the revised manuscript to include mean and standard deviation values for all reported results.
>
> > **S2: The description of the proposed method is sometimes unclear, and several components would benefit from improved clarity and organization.**
>
> **R**: We thank the reviewer for this comment and appreciate the suggestion to improve the clarity and organization of the method description. We aim to make the manuscript as clear and accessible as possible. Since the comment does not specify which parts were unclear, we would greatly appreciate more detailed feedback or examples of sections that were difficult to follow. This will help us better target the revisions.
>
>
> # Reply to Question
>
> > **Q1:If I understand correctly, each decoding task is said to have its own set of classification tokens within this multi-task model. How does the model handle new, unseen tasks during inference? Is there a mechanism for extending the task token space without retraining the entire model?**
>
> **R**: Yes, each decoding task in our framework is associated with its own set of classification tokens. As with any standard classification model, presenting an out-of-distribution task at inference without adaptation is not specific to neural decoding. The scenario described by the reviewer aligns with an incremental learning setting, in which new classes or tasks are introduced and the model is adapted without full retraining. In principle, a new classification token could be added for the unseen task, and the model could be fine-tuned by following established practices in the incremental learning literature to incorporate the new decoding objective. Exploring such extensions is a promising direction for future work.

---

> ### Author Response · Authors · 2025-11-14
> **Response to Reviewer ebUX (3/3)**
>
> > **Q2: When comparing against baselines such as PopT that are also trained on intracranial recordings, are these baselines fine-tuned from their pre-trained weights, or trained from scratch?**
>
> **R**: We thank the reviewer for this question. All baselines were reproduced by strictly following the official implementations and configurations provided by the original authors. For methods involving pre‑training, we re‑pretrained the models on our dataset before performing decoding.
>
> In the case of PopT, we hypothesize that its lower performance in our setting may be due to input feature incompatibility for language decoding in speech production. PopT employs STFT‑based spectral features, which may be advantageous for its original decoding tasks-primarily speech perception (listening and reading) and distinguishing pitch, volume, and speech vs. non‑speech—as these tasks can benefit from frequency‑domain representations and carefully tuned STFT parameters.
>
> However, our decoding task focuses on speech production, which involves fine orofacial and articulatory muscle movements. Capturing these subtle temporal patterns may be better served by using raw time‑domain neural signals as input, as is standard practice in recent high‑performance speech decoding studies [1,2]. This also suggest that our model is more generic.
>
> # References
>
> [1] Metzger S L, Littlejohn K T, Silva A B, et al. A high-performance neuroprosthesis for speech decoding and avatar control[J]. Nature, 2023, 620(7976): 1037-1046.
>
> [2] Willett F R, Kunz E M, Fan C, et al. A high-performance speech neuroprosthesis[J]. Nature, 2023, 620(7976): 1031-1036.
>
> > **Q3: Given that prior works such as PopT have already tackled the challenge of inter-subject electrode placement variability, what is the key novelty or improvement introduced by the proposed method in this aspect?**
>
> **R**:  Our study specifically targets **multi-task, multi-subject** intracranial decoding, whereas PopT addresses only **single task** decoding. Furthermore, Neuro-MoBRE is designed to tackle multiple forms of heterogeneity beyond electrode implantation differences. The four core components target distinct sources of variability:
> * Brain-regional-temporal tokenizer and regional masked autoencoding primarily address multi-subject heterogeneity.
> * Brain-regional MoE and Task-disentangled Information Aggregation primarily address multi-task heterogeneity.
>
> While PopT primarily focuses on variability from electrode placement, our framework simultaneously addresses both multi-subject and multi-task heterogeneity, providing a more general and extensible decoding solution.
>
> > **Q4: What distinguishes the proposed brain-regional MoE from a standard MoE architecture (from the architecture aspect)?**
>
> **R**: From an algorithmic perspective, a Mixture-of-Experts (MoE) architecture can be broadly characterized by (i) the design of its routing mechanism, and (ii) the structure of its experts [1].
>
> In a standard MoE, each token is independently routed—based on its own features—to one or more experts. In contrast, our brain‑regional MoE uses a routing mechanism that **assigns all tokens originating from the same brain region to the same expert**. This design ensures that each expert specializes in processing neural features from a specific brain region, promoting region‑specific representation learning and aligning the model architecture more closely with the underlying neuroanatomy.
>
> # References
>
> [1] Cai, Weilin, et al. "A survey on mixture of experts in large language models." IEEE Transactions on Knowledge and Data Engineering (2025).
>
> > **Q5: Will the dataset be released publicly if the paper is accepted?**
>
> **R**: We will make the dataset available upon reasonable request to the corresponding authors, in accordance with all applicable ethical and privacy regulations.

---

> ### Author Response · Authors · 2025-11-24
>
> Dear Reviewer ebUX,
>
> Thank you for your thoughtful review and for raising important points regarding our work. We have provided detailed responses to each of your comments in the rebuttal.
>
> We would greatly appreciate it if you could revisit our responses when you have time. Please let us know if any concerns remain or if further clarification would help. If you feel that our rebuttal satisfactorily addresses your questions, we kindly ask you to consider reflecting this in your overall score.
>
> We value your feedback and willingness to engage in the discussion, and we thank you again for helping us improve the manuscript.
>
> Authors

---

> > ### Author Response · Authors · 2025-11-27
> > **Follow-Up on Rebuttal Response**
> >
> > Dear Reviewer ebUX,
> >
> > We are writing to kindly follow up on our responses to your insightful comments and feedback. We have provided detailed clarifications and additional analyses addressing your concerns and suggestions in our rebuttal.
> >
> > If there are any remaining questions or points you'd like us to address further, please do not hesitate to let us know. Your feedback has been invaluable in refining our work, and we would greatly appreciate it if you could share your thoughts on whether our responses meet your expectations.
> >
> > Thank you once again for your time and effort in reviewing our submission.
> >
> > Authors

---

### Official Review · Reviewer_egsk · 2025-10-30

**Soundness:** 2
**Presentation:** 2
**Contribution:** 2
**Rating:** 2
**Confidence:** 4

**Summary:**

This paper introduces Neuro-MoBRE, a novel neural decoding framework designed to explicitly resolve the pervasive data heterogeneity in multi-subject, multi-task intracranial neurophysiological decoding. The framework integrates a brain-regional-temporal embedding mechanism, a Mixture of Brain Regional Experts (BrMoE) module, and a task-disentangled information aggregation mechanism. It is further enhanced by a region-masked autoencoding (RMAE) pre-training strategy to improve generalization. Evaluated on intracranial recordings from 11 subjects across five diverse tasks, including language decoding and seizure diagnosis, Neuro-MoBRE demonstrates superior performance over existing methods.

**Strengths:**

1. Well-Motivated and Novel Methodology: The paper accurately identifies "data heterogeneity" as a fundamental challenge in neural decoding and proposes a systematic solution. The core ideas, particularly the brain-regional MoE and task-disentangled aggregation, are innovative.

2. Multi-Subject, Multi-Task Modeling: The framework successfully unifies data from multiple subjects and tasks within a single model. Its demonstrated zero-shot generalization to unseen subjects have some advantages with practical value.

3. Rigorous Experimental Design: The evaluation covering challenging tasks like Mandarin phonological decoding (initials, finals, tones) and clinical epilepsy diagnosis (detection and prediction) using real sEEG data, making the results credible.

4. Modular Design and Thorough Ablation Studies: The model is decomposed into key components (BrMoE, TIA, RMAE), and extensive ablation studies are conducted to validate the contribution of each, solidifying the methodological claims.

**Weaknesses:**

1. Limited Generalization Evidence: The model is evaluated solely on one private iEEG dataset. Its generalization capability remains unverified on any public iEEG benchmarks with varying experimental paradigms and recording parameters [1,2], raising questions about its robustness across broader data distributions.

2. Brain Region Modeling: The current approach models neural activity at the level of entire brain regions, potentially overlooking the functional complexity and finer-grained functional sub-divisions or dynamic network interactions within these regions.

3. Limited Applicability: While the framework is innovative, the absolute performance for language decoding remains low (e.g., ~29% top-1 accuracy for initial decoding in Table 3). This level of accuracy is far from sufficient for practical clinical applications in assistive communication, highlighting a significant gap towards immediate real-world impact.

4. Lacks In-Depth Comparison with State-of-the-Art Baselines: Comparisons with recent foundational models for neural decoding [3,4] are relatively limited. The paper does not fully establish its superior advantage in unified multi-task modeling against these strong contenders.

**References:**

[1] Wang, C., Yaari, A., Singh, A., Subramaniam, V., Rosenfarb, D., DeWitt, J., ... & Barbu, A. (2024). Brain treebank: Large-scale intracranial recordings from naturalistic language stimuli. *Advances in Neural Information Processing Systems, 37*, 96505-96540.

[2] Zheng, H., Wang, H., Jiang, W., Chen, Z., He, L., Lin, P., ... & Liu, Y. (2024). Du-IN: Discrete units-guided mask modeling for decoding speech from Intracranial Neural signals. *Advances in Neural Information Processing Systems, 37*, 79996-80033.

[3] Singh, A., Thomas, T., Li, J., Hickok, G., Pitkow, X., & Tandon, N. (2025). Transfer learning via distributed brain recordings enables reliable speech decoding. *Nature Communications, 16*(1), 8749.

[4] Chen, X., Wang, R., Khalilian-Gourtani, A., Yu, L., Dugan, P., Friedman, D., ... & Flinker, A. (2024). A neural speech decoding framework leveraging deep learning and speech synthesis. *Nature Machine Intelligence, 6*(4), 467-480.

**Questions:**

See Weaknesses.

**Details Of Ethics Concerns:**

While the authors note IRB approval and informed consent, the paper falls short in several aspects of responsible research practice, which are critical for studies involving sensitive intracranial data from a clinical population. Firstly, there is no mention of any plan to release the dataset or make it accessible to the research community, severely hindering reproducibility and the broader impact of this work. Secondly, the scope of the informed consent is unclear, particularly regarding future use in large-scale model training and data sharing. Thirdly, beyond basic anonymization, there is no discussion of additional measures to protect participant privacy against potential re-identification risks. I strongly encourage the authors to address these points, ideally by committing to a data release strategy and elaborating on the ethical safeguards implemented.

---

> ### Author Response · Authors · 2025-11-14
> **Response to Reviewer egsk (1/3)**
>
> ## Response to Reviewer egsk
> ---
>
> We thank the reviewer for the thoughtful assessment and for recognizing several key strengths of our work, including its well‑motivated and novel contributions with practical value, rigorous experimental design with thorough ablation studies. While we are encouraged by the positive remarks, we were somewhat disappointed to see that these strengths were not reflected in the overall score.
>
> # Reply to Weakness
>
> > **W1: Limited Generalization Evidence: The model is evaluated solely on one private iEEG dataset. Its generalization capability remains unverified on any public iEEG benchmarks with varying experimental paradigms and recording parameters [1,2], raising questions about its robustness across broader data distributions.**
>
> **R**: Our study specifically targets **multi-task, multi-subject** decoding, for which no suitable public datasets currently exist.  Merging multiple publicly available datasets [1,2] often introduces substantial variability due to differences in equipment, acquisition protocols, and subject cohorts, which can confound the evaluation of methods for handling task-related and subject-specific heterogeneity. In our early investigations, such heterogeneity significantly degraded decoding performance.
>
> Compared to decoding tasks such as detecting abnormal EEG states, tasks that are relatively with minimal cognitive demands and often separable by amplitude or frequency features. Our work addresses higher-level cognitive and semantic decoding tasks, such as phoneme classification with 23 distinct classes, which exhibit greater inter-subject heterogeneity.
>
> To mitigate unnecessary heterogeneity and enable a controlled demonstration, we collected intracranial recordings from the same participant cohort using identical equipment and protocols, and conducted five distinct decoding tasks for each of 11 participants, including phoneme-level language decoding and epileptic seizure diagnosis. To the best of our knowledge, this represents the first multi-task, multi-subject intracranial decoding study, and we will release the dataset publicly upon acceptance.
>
> # References
>
> [1]  Wang, C., Yaari, A., Singh, A., Subramaniam, V., Rosenfarb, D., DeWitt, J., ... & Barbu, A. (2024). Brain treebank: Large-scale intracranial recordings from naturalistic language stimuli. Advances in Neural Information Processing Systems, 37, 96505-96540.
>
> [2] Zheng, H., Wang, H., Jiang, W., Chen, Z., He, L., Lin, P., ... & Liu, Y. (2024). Du-IN: Discrete units-guided mask modeling for decoding speech from Intracranial Neural signals. Advances in Neural Information Processing Systems, 37, 79996-80033.
>
> > **W2: Brain Region Modeling: The current approach models neural activity at the level of entire brain regions, potentially overlooking the functional complexity and finer-grained functional sub-divisions or dynamic network interactions within these regions.**
>
> **R**: We thank the reviewer for raising this point. Studying finer-grained functional subdivisions or dynamic network interactions within brain regions would **require high-density ECoG or MEA arrays implanted across multiple regions**, which is currently **not feasible in clinical studies** based on current recording techniques. We emphasize that these spatial-resolution limitations are a well-known challenge shared by the broader neuroscience and brain-computer interface fields, and are not unique to our work.
>
> Existing intracranial decoding work [1,2,3] typically focuses on single-task decoding within a single subject, often with electrodes covering only one brain region. In our study, we collected sEEG recordings from 11 participants performing five distinct tasks. Because sEEG provides spatial samples at selected sites rather than continuous coverage of an entire region, our modeling is done at the brain-region level. This approach reflects the best resolution attainable given current sEEG implantation practices, especially when working across many subjects and tasks. Our contribution lies in developing a general invasive multi-subject, multi-task decoding framework that explicitly addresses heterogeneity under realistic clinical constraints.
>
> # References
>
> [1] Metzger S L, Littlejohn K T, Silva A B, et al. A high-performance neuroprosthesis for speech decoding and avatar control[J]. Nature, 2023, 620(7976): 1037-1046.
>
> [2] Willett F R, Kunz E M, Fan C, et al. A high-performance speech neuroprosthesis[J]. Nature, 2023, 620(7976): 1031-1036.
>
> [3] Bouton, Chad E., et al. "Restoring cortical control of functional movement in a human with quadriplegia." Nature 533.7602 (2016): 247-250.

---

> ### Author Response · Authors · 2025-11-14
> **Response to Reviewer egsk (2/3)**
>
> > **W3: Limited Applicability: While the framework is innovative, the absolute performance for language decoding remains low (e.g., ~29% top-1 accuracy for initial decoding in Table 3). This level of accuracy is far from sufficient for practical clinical applications in assistive communication, highlighting a significant gap towards immediate real-world impact.**
>
> **R**: We appreciate the reviewer's recognition of the framework's novelty. Our primary objective is to introduce a **generic multi-task, multi-subject intracranial decoding framework** that explicitly addresses the heterogeneity arising from biological differences across subjects, as well as variability in experimental setups and conditions rather than to optimize solely for immediate clinical deployment in language decoding.
>
> Regarding the reported top-1 accuracy (~29% in Table 3), we note that this is the average across 11 subjects. Performance varies substantially due to individual heterogeneity and dialectal pronunciation habits, with the highest accuracy reaching ~57% and the lowest around ~15%. For the participant with the lowest accuracy, distinctive regional dialect patterns (e.g., frequent confusion between 'n' and 'l', and between alveolar 's' and retroflex 'sh') reduced discriminability at the neural level. Conversely, participants with higher performance achieved results consistent with state-of-the-art phoneme decoding reported in prior works [1,2], which could be further integrated with language model based post-processing to enable deployment as a speech neuroprosthesis.
>
> # References
>
> [1] Metzger S L, Littlejohn K T, Silva A B, et al. A high-performance neuroprosthesis for speech decoding and avatar control[J]. Nature, 2023, 620(7976): 1037-1046.
>
> [2] Willett F R, Kunz E M, Fan C, et al. A high-performance speech neuroprosthesis[J]. Nature, 2023, 620(7976): 1031-1036.
>
>
> > **W4: Lacks In-Depth Comparison with State-of-the-Art Baselines: Comparisons with recent foundational models for neural decoding [1,2] are relatively limited. The paper does not fully establish its superior advantage in unified multi-task modeling against these strong contenders.**
>
> **R**: We respectfully disagree. Our study specifically targets multi-task, multi-subject **intracranial decoding**,  whereas the two referenced works address different problem settings.
>
> In particular, Singh et al. propose a cross-subject transfer learning framework, not a unified multi-task, multi-subject approach. Furthermore, Singh et al. was published in October
> 2025, well after the submission of our manuscript, making direct comparison infeasible at the time of submission.
>
> Chen et al. focus on single-subject speech decoding from intracranial recordings, which does not address multi-task modeling or the heterogeneity challenges that arise across subjects.
>
> While these methods are valuable within their respective problem domains, their objectives and experimental settings differ substantially from ours.
>
> # References
>
> [1] Singh, A., Thomas, T., Li, J., Hickok, G., Pitkow, X., & Tandon, N. (2025). Transfer learning via distributed brain recordings enables reliable speech decoding. Nature Communications, 16(1), 8749.
>
> [2] Chen, X., Wang, R., Khalilian-Gourtani, A., Yu, L., Dugan, P., Friedman, D., ... & Flinker, A. (2024). A neural speech decoding framework leveraging deep learning and speech synthesis. Nature Machine Intelligence, 6(4), 467-480.

---

> ### Author Response · Authors · 2025-11-14
> **Response to Reviewer egsk (3/3)**
>
> # Reply to Ethics Concerns
>
> > **While the authors note IRB approval and informed consent, the paper falls short in several aspects of responsible research practice, which are critical for studies involving sensitive intracranial data from a clinical population. Firstly, there is no mention of any plan to release the dataset or make it accessible to the research community, severely hindering reproducibility and the broader impact of this work. Secondly, the scope of the informed consent is unclear, particularly regarding future use in large-scale model training and data sharing. Thirdly, beyond basic anonymization, there is no discussion of additional measures to protect participant privacy against potential re-identification risks. I strongly encourage the authors to address these points, ideally by committing to a data release strategy and elaborating on the ethical safeguards implemented.**
>
> **R**: We thank the reviewer for emphasizing the importance of responsible research practices:
>
> * **Informed Consent**: We will release the dataset upon acceptance of this paper, subject to standard data use agreements to ensure ethical and legal compliance. Access will be provided to qualified researchers upon reasonable request.
>
> * **Dataset Release**: All participants provided written informed consent approved by our IRB, which explicitly covers research use and sharing of their anonymized data.
>
> * **Data anonymization**: In addition to standard anonymization (removal of direct identifiers and metadata scrubbing), we also ensure all data is stripped of any session-specific meta-information (e.g., timestamps tied to hospital records).
>
> We will add a dedicated section in the revised manuscript to to reflect these points.

---

> ### Author Response · Authors · 2025-11-24
>
> Dear Reviewer egsk,
>
> Thank you for your thoughtful review and for raising important points regarding our work. We have provided detailed responses to each of your comments in the rebuttal.
>
> We would greatly appreciate it if you could revisit our responses when you have time. Please let us know if any concerns remain or if further clarification would help. If you feel that our rebuttal satisfactorily addresses your questions, we kindly ask you to consider reflecting this in your overall score.
>
> We value your feedback and willingness to engage in the discussion, and we thank you again for helping us improve the manuscript.
>
> Authors

---

> > ### Author Response · Authors · 2025-11-27
> > **Follow-Up on Rebuttal Response**
> >
> > Dear Reviewer egsk,
> >
> > Thank you again for your thoughtful review and for acknowledging the novelty and practical value of our work.
> >
> > We would greatly appreciate it if you could revisit our responses to see whether they satisfactorily address your earlier questions. If any points remain unclear, we would be happy to provide further explanation. Should the clarifications resolve your concerns, we kindly ask you to consider updating your evaluation accordingly.
> >
> > Thank you once again for your time and valuable input.
> >
> > Authors

---

### Official Review · Reviewer_ZzKV · 2025-11-01

**Soundness:** 3
**Presentation:** 2
**Contribution:** 2
**Rating:** 4
**Confidence:** 4

**Summary:**

The paper introduces a general-purpose neural decoding framework designed to handle the pervasive heterogeneity in multi-subject, multi-task intracranial recordings (ECoG/sEEG). The motivation stems from the limitations of existing neurophysiological decoding models, which often perform well only in single-task or single-subject settings and fail to generalize across heterogeneous datasets. The proposed framework, Neuro-MoBRE, explicitly models the variability across brain regions, subjects, and tasks to achieve robust generalization in brain decoding. Evaluations on intracranial recordings from subjects across diverse tasks show the performance of the proposed framework.

**Strengths:**

1. The paper presents a framework to address cross-subject and cross-task heterogeneity. By using a brain-regional mixture-of-experts mechanism, a brain-regional-temporal tokenizer, and task-disentangled aggregation, the framework separates regional, temporal, and functional variability of EEG data.

2. The authors curate and unify one of the most comprehensive intracranial EEG datasets to date, spanning 11 subjects and five heterogeneous tasks, including speech decoding, movement execution, and epileptic activity classification.

3. The authors benchmark Neuro-MoBRE against a range of recent baselines (e.g., BIOT, LaBraM, NeuroLM) and demonstrate consistent improvements in accuracy and cross-subject generalization.

**Weaknesses:**

1. While the paper emphasizes that Neuro-MoBRE is designed to “explicitly resolve multi-subject and multi-task heterogeneity,” the empirical evidence for this claim is qualitative and indirect. The results show performance gains across subjects and tasks, but it remains unclear how much of that improvement is attributable to reduced heterogeneity versus general over-parameterization or better representation learning.

2. The paper claims robustness to low-SNR neural recordings, but the evidence remains unquantified. Although masked pretraining and expert specialization are helpful, there is no experiment showing that the model’s performance degrades less severely under noisy or limited-channel conditions than baselines.

3. The evaluation focuses on accuracy and ablation gains but provides limited insight into the learned representations or biological plausibility. There is no quantitative measure of region-expert correspondence, no evaluation of representational disentanglement.

4. The references to Figure A1 and Table A1 in lines 312 and 313 are incorrect. They actually refer to Figure 1 and Table 1, not the ones in the appendix.

**Questions:**

1. How can the authors demonstrate that Neuro-MoBRE truly resolves heterogeneity across subjects and tasks, rather than simply benefiting from larger capacity or better feature sharing? Would quantitative analyses such as inter-subject representational similarity, variance reduction, or expert-routing ablations help substantiate this claim?

2. Can the authors provide evidence on solving the low-SNR issue? Like the robustness under low-SNR conditions?

3. What interpretability or representational analyses can clarify what each regional expert learns?

---

> ### Author Response · Authors · 2025-11-14
> **Response to Reviewer ZzKV (1/2)**
>
> ## Response to Reviewer ZzKV
> ---
>
> # Reply to Weakness
>
> > **W1: While the paper emphasizes that Neuro-MoBRE is designed to 'explicitly resolve multi-subject and multi-task heterogeneity' the empirical evidence for this claim is qualitative and indirect. The results show performance gains across subjects and tasks, but it remains unclear how much of that improvement is attributable to reduced heterogeneity versus general over-parameterization or better representation learning.**
>
> **R**: We thank the reviewer for raising this important point. Our model is explicitly designed to handle multi-subject and multi-task heterogeneity, and we view **resolving such heterogeneity as a fundamental step toward learning better neural representations**. The four core components of our framework target different sources of heterogeneity in this setting:
> * Brain-regional-temporal tokenizer and regional masked autoencoding primarily address multi-subject heterogeneity.
> * Brain-regional MoE and Task-disentangled Information Aggregation primarily address multi-task heterogeneity.
>
> To quantify their contribution, we present ablation results in Table 3 of the manuscript. Starting from a base model, the averaged top-1 accuracy increases from ~18% to ~29% when all heterogeneity-handling components are included, demonstrating that each module contributes significantly to improved performance and higher-quality learned representations.
>
> We also compared the parameter count of Neuro-MoBRE to existing baselines and found that our model has fewer parameters. This rules out over-parameterization as the primary reason for the observed gains, and supports our conclusion that the improvements are due to the effective resolution of heterogeneity rather than simply higher capacity.
>
>  |       Method      | # Param |
> |:-----------------:|:-------:|
> |       Brant       |   500M  |
> |       LaBraM      |   350M  |
> |        BIOT       |    3M   |
> |       PopT        |   20M   |
> | NEURO-MOBRE(Ours) |    5M   |
>
>
> > **W2: The paper claims robustness to low-SNR neural recordings, but the evidence remains unquantified. Although masked pretraining and expert specialization are helpful, there is no experiment showing that the model's performance degrades less severely under noisy or limited-channel conditions than baselines.**
>
> **R**: We thank the reviewer for this comment and would like to clarify a potential misunderstanding of our claim regarding low-SNR recordings. In this work, we do not aim to directly solve the low-SNR problem in non-invasive neurophysiological recordings. Rather, we aim to **circumvent these limitations by collecting intracranial recordings** from multiple subjects across distinct decoding tasks, which inherently benefit from higher SNR compared to non-invasive modalities.
>
> We note that the low SNR of EEG, due to signal attenuation from the skull and dura mater, exacerbates heterogeneity in higher-level cognitive and semantic decoding tasks (e.g., the language decoding examined in our study). To mitigate such unnecessary heterogeneity, we collected intracranial recordings from the same participant cohort, using identical equipment and protocols, and carried out five distinct decoding tasks (including phoneme-level language decoding and epileptic seizure diagnosis) for each of 11 participants.
>
> > **W3: The evaluation focuses on accuracy and ablation gains but provides limited insight into the learned representations or biological plausibility. There is no quantitative measure of region-expert correspondence, no evaluation of representational disentanglement.**
>
> **R**: We thank the reviewer for this comment. To quantify the correspondence between brain regions and experts, we analyzed and visualized the distribution of tokens from each region as processed by each expert in our Brain-regional MoE module. This provides a **quantitative measurement** of region-expert specialization.
>
> As shown in Fig. 4, token distributions differ substantially across experts, indicating specialized functional roles. For example, Expert 1 predominantly processes neural signals from the superior temporal gyrus (STG), which is well-known to be involved in auditory processing for speech and language. Experts 2, 8 and 13 mainly process signals from the pre-central and post-central gyrus, regions associated with speech production and motor coordination of speech movements.
>
> This analysis binds experts to region-specific representations, revealing distinct expert preferences. With prior knowledge of each brain region's function, we can infer the likely role of each expert in processing specific neural functions. We hypothesize that, with additional decoding tasks, these experts could be interpreted as modeling functionally distinct **neural circuits or subnetworks**, thereby providing both representational disentanglement and biological plausibility.

---

> ### Author Response · Authors · 2025-11-14
> **Response to Reviewer ZzKV (2/2)**
>
> > **W4: The references to Figure A1 and Table A1 in lines 312 and 313 are incorrect. They actually refer to Figure 1 and Table 1, not the ones in the appendix.**
>
> **R**: We thank the reviewer for pointing out the incorrect links to Figure A1 and Table A1 on page 6. We have corrected the links in the revised manuscript to point to the intended figure and table.
>
>
> # Reply to Question
>
> > **Q1:How can the authors demonstrate that Neuro-MoBRE truly resolves heterogeneity across subjects and tasks, rather than simply benefiting from larger capacity or better feature sharing? Would quantitative analyses such as inter-subject representational similarity, variance reduction, or expert-routing ablations help substantiate this claim?**
>
> **R**: We thank the reviewer for this question and the concrete suggestions. As noted in our response to Weakness 1, our ablation studies demonstrate that each of the four proposed components effectively addresses different aspects of heterogeneity, and that performance gains are not due to over-parameterization.
>
> To further quantify the impact on inter-subject heterogeneity, we measured the Euclidean distance between participant-specific representation centroids in the feature space (immediately before the decoding head) under two training regimes:
>
> * Single-subject setting: three separate models trained independently for participants 1, 10, and 11.
> * Multi-subject setting: a single Neuro-MoBRE model trained jointly on all subjects.
>
> We report distances between subject 1 and the other two:
>
> | Subject | Single-sub | Multi-sub |
> |:-------:|:----------:|-----------|
> |  1<->10 |  328305.81 | 7782      |
> |  1<->11 |  228875.05 | 5860      |
>
> The substantially smaller distances in the multi-subject setting indicate greater alignment of the learned feature space across participants, suggesting that Neuro-MoBRE indeed reduces inter-subject representational disparity, a direct sign of heterogeneity mitigation. This analysis complements our existing ablations and provides quantitative evidence that the observed gains stem from heterogeneity resolution rather than only larger capacity or generic feature sharing.
>
> > **Q2: Can the authors provide evidence on solving the low-SNR issue? Like the robustness under low-SNR conditions?**
>
> **R**: Please refer to our response to Weakness two.
>
> > **Q3: What interpretability or representational analyses can clarify what each regional expert learns?**
>
> **R**: Please refer to our response to Weakness three.

---

> ### Author Response · Authors · 2025-11-24
>
> Dear Reviewer ZzKV,
>
> Thank you for your thoughtful review and for raising important points regarding our work. We have provided detailed responses to each of your comments in the rebuttal.
>
> We would greatly appreciate it if you could revisit our responses when you have time. Please let us know if any concerns remain or if further clarification would help. If you feel that our rebuttal satisfactorily addresses your questions, we kindly ask you to consider reflecting this in your overall score.
>
> We value your feedback and willingness to engage in the discussion, and we thank you again for helping us improve the manuscript.
>
> Authors

---

> > ### Author Response · Authors · 2025-11-27
> > **Follow-Up on Rebuttal Response**
> >
> > Dear Reviewer ZzKV,
> >
> > Thank you again for your thoughtful review and for acknowledging the novelty and practical value of our work.
> >
> > We would greatly appreciate it if you could revisit our responses to see whether they satisfactorily address your earlier questions. If any points remain unclear, we would be happy to provide further explanation. Should the clarifications resolve your concerns, we kindly ask you to consider updating your evaluation accordingly.
> >
> > Thank you once again for your time and valuable input.
> >
> > Authors

---

### Author Response · Authors · 2025-11-25
**General Response (1/2)**

We thank all reviewers for their time, effort, and constructive feedback. We are encouraged that the reviews acknowledge the novelty, significance, and contributions of our work; however, we note that these positive assessments are not fully reflected in the overall scores. We hope that our detailed responses, clarifications, and additional experiments will help address any remaining concerns and lead to a reevaluation of the paper's merit.

### Recap of Our Contribution
We would like to briefly restate the motivation and main contributions of our work:

* **First generic multi-task, multi-subject intracranial decoding framework:**
  To the best of our knowledge, this is the first framework that supports **multi-task neural decoding** across multiple subjects without requiring fine-tuning for each single-task dataset. In contrast, existing models such as LaBraM [1], although pre-trained on multiple datasets, require task-specific fine-tuning for single-task downstream decoding.

* **Systematic approach to resolving data heterogeneity:**
  We identify **data heterogeneity**—arising from biological differences, acquisition setups, and task variability—as a fundamental challenge in neural decoding, and propose a systematic architectural solution to address it.

* **Curated comprehensive intracranial EEG dataset:**
  We collected a dataset covering **five distinct tasks** from the **same cohort of participants**, enabling controlled **multi-task** and **multi-subject** neural decoding evaluations. We will make this dataset publicly available upon acceptance.

---

### Clarification of Common Misunderstandings

#### **1. Choice of curating our own intracranial multi-subject, multi-task dataset**
NeuroLM [2], another multi-task model, is conceptually similar to Neuro‑MoBRE. However, the inferior performance of NeuroLM compared to LaBraM on TUAB and TUEV—a drop of roughly **20% in balanced accuracy**—suggests that **heterogeneity across current EEG datasets** (differences in equipment, acquisition protocols, and subject cohorts) makes true multi-task decoding difficult when merging existing public EEG datasets. This is a key reason we curated a controlled, homogeneous intracranial dataset for our study.

We further conducted experiments on TUAB and TUEV comparing Neuro‑MoBRE, NeuroLM, and LaBraM.

**Performance on TUAB**

| Method        | Multi‑task | Balanced Acc      | AUROC             |
|:-------------:|:----------:|:-----------------:|:-----------------:|
| LaBraM [1]    | No         | 0.8140 ± 0.0019   | 0.9022 ± 0.0009   |
| NeuroLM [2]   | Yes        | 0.7826 ± 0.0065   | 0.6975 ± 0.0081   |
| **Neuro‑MoBRE** | Yes        | **0.8199 ± 0.0056** | **0.8873 ± 0.0062** |

**Performance on TUEV**

| Method        | Multi‑task | Balanced Acc      | Cohen’s Kappa     |
|:-------------:|:----------:|:-----------------:|:-----------------:|
| LaBraM [1]    | No         | 0.6409 ± 0.0065   | 0.6637 ± 0.0093   |
| NeuroLM [2]   | Yes        | 0.4560 ± 0.0048   | 0.4285 ± 0.0048   |
| **Neuro‑MoBRE** | Yes        | **0.5580 ± 0.0076** | **0.5428 ± 0.0089** |

From these results:
- NeuroLM (multi-task) performs **significantly worse** than LaBraM (single-task), particularly on TUEV (~18% drop in balanced accuracy).
- Neuro‑MoBRE outperforms NeuroLM by a large margin on both datasets, demonstrating **better heterogeneity handling** despite still trailing LaBraM.

These findings reinforce our position that merging heterogeneous EEG datasets limits the potential of multi-task frameworks, and that a curated, controlled dataset—like the one used in our main experiments—is essential for fair and effective multi-task evaluation.

---

---

> ### Author Response · Authors · 2025-11-25
> **General Response (2/2)**
>
> #### **2. Neuro‑MoBRE is a generic decoding framework, not a speech prosthesis**
> The primary objective of our work is to propose a **generic multi-task, multi-subject intracranial decoding framework** that explicitly addresses heterogeneity arising from biological differences across subjects, as well as variability in experimental setups and conditions. Our contribution centers on a general decoding architecture—not solely sentence-level language decoding.
>
> To demonstrate the flexibility of the framework, we used phoneme-level decoding tasks and seizure diagnostic tasks as proof-of-concept applications. While many non-invasive language decoding approaches (e.g., EEG [3] and fMRI [4]) achieve sentence-level decoding, they often rely on **retrieval-based methods** using pre-trained large language models (LLMs), where neural representations are mapped into an LLM embedding space and decoded sentences are retrieved.
>
> By contrast, we perform **direct phoneme-level decoding**, following recent state-of-the-art approaches such as Metzger et al. [5] and Willett et al. [6]. From decoded phoneme sequences, sentence-level decoding can readily be achieved using language model post-processing (e.g., n‑gram models), but implementing and evaluating these pipelines would require substantial engineering effort beyond the scope of the current work. Nevertheless, our phoneme decoding results demonstrate both the **effectiveness** and **extensibility** of the proposed framework.
>
> ---
>
> **References**
> [1] Jiang, W.-B., Zhao, L.-M., & Lu, B.-L. *Large brain model for learning generic representations with tremendous EEG data in BCI*. arXiv:2405.18765 (2024).
>
> [2] Jiang, W., Wang, Y., Lu, B.-L., & Li, D. *NeuroLM: A universal multi-task foundation model for bridging the gap between language and EEG signals*. ICLR 2025.
>
> [3] Liu, H., et al. *EEG2Text: Open vocabulary EEG-to-text decoding with EEG pre-training and multi-view transformer*. arXiv:2405.02165 (2024).
>
> [4] Qiu, W., et al. *MindLLM: A subject-agnostic and versatile model for fMRI-to-text decoding*. arXiv:2502.15786 (2025).
>
> [5] Metzger, S.L., Littlejohn, K.T., Silva, A.B., et al. *A high-performance neuroprosthesis for speech decoding and avatar control*. Nature, 620(7976): 1037–1046 (2023).
>
> [6] Willett, F.R., Kunz, E.M., Fan, C., et al. *A high-performance speech neuroprosthesis*. Nature, 620(7976): 1031–1036 (2023).
>
> ---

---

### Author Response · Authors · 2025-11-27
**Follow-Up on Rebuttal Response**

We thank all reviewers again for their time, effort, and constructive feedback. Since there has been no discussion so far, we would like to kindly invite you to revisit our detailed rebuttal. In our responses, we clarified key points, addressed each concern raised, and provided additional experiments, analyses, and manuscript revisions to strengthen the work.

If any questions or uncertainties remain, we would be happy to provide further clarification or additional supporting material.
If you feel that our rebuttal satisfactorily addresses your comments, we would appreciate it if you could consider reflecting that in your overall assessment.

We value your engagement in this process and look forward to any further discussion before the review period concludes.

---

### Meta-Review · Area_Chair_dhFu · 2026-01-06

**Summary:**

The paper proposes Neuro-MoBRE, a decoder-only transformer with brain-regional–temporal embeddings designed to address heterogeneity and low SNR in intracranial neurophysiological decoding, and evaluates on intracranial recordings from 11 subjects across five tasks.

While the framing is interesting and the architecture aims at an important challenge, the reviews are uniformly negative due to substantial gaps in validation and positioning. A major concern is the lack of experiments on publicly available benchmarks, which makes it difficult to contextualize progress against closely related works. Reviewers also note that the paper appears to overclaim zero-shot generalization, and that the unseen-subject setting is not evaluated against strong baselines, limiting confidence in the proposed approach’s generalization claims. More broadly, the empirical comparisons to state-of-the-art methods are not sufficiently comprehensive, and the reported absolute performance is said to be still far from what would be needed for practical assistive or clinical deployment, weakening the claimed real-world impact. Given these issues around benchmark coverage, overstatement of generalization, missing baseline comparisons, and limited demonstrated applicability, it is not good for ICLR.

**Reviewer Concerns:**

.

**Reviewer Scores:**

.

---

### Decision · Program_Chairs · 2026-01-26

Reject